# Statistically Meaningful Approximation: a Case Study on Approximating Turing Machines with Transformers

**Colin Wei**    **Yining Chen**    **Tengyu Ma**

Department of Computer Science
Stanford University

{colinwei,cynnjjs,tengyuma}@cs.stanford.edu

## Abstract

A common lens to theoretically study neural net architectures is to analyze the functions they can approximate. However, the constructions from approximation theory often have unrealistic aspects, for example, reliance on infinite precision to memorize target function values. To address this issue, we propose a formal definition of statistically meaningful approximation which requires the approximating network to exhibit good statistical learnability. We present case studies on statistically meaningful approximation for two classes of functions: boolean circuits and Turing machines. We show that overparameterized feedforward neural nets can statistically meaningfully approximate boolean circuits with sample complexity depending only polynomially on the circuit size, not the size of the approximating network. In addition, we show that transformers can statistically meaningfully approximate Turing machines with computation time bounded by $T$, requiring sample complexity polynomial in the alphabet size, state space size, and $\log(T)$. Our analysis introduces new tools for generalization bounds that provide much tighter sample complexity guarantees than the typical VC-dimension or norm-based bounds, which may be of independent interest.

## 1   Introduction

Dating back to the seminal works on universal approximation [16, 25, 40, 31], a common way to theoretically study neural nets has been through their expressivity, which measures the ability of neural nets to approximate well-behaved functions. This perspective has shaped how researchers perceive different types of deep learning architectures: a basic way to theoretically justify new architectures is to study their approximation capabilities. This has led to a number of analyses studying universal approximation capabilities for various widely-used architectures, such as recurrent neural nets (RNNs) [47], graph neural nets [46], convolutional networks [3, 64, 59], residual networks [32], transformers [61], and neural ODEs [51, 63].

However, approximation theoretic results often misalign with more meaningful end-to-end guarantees, because models constructed in the literature often exhibit unrealistic properties. For example, a common technique in the universal approximation literature is to rely strongly on infinite-precision weights and activations, or exponentially many parameters to encode the desired function values [25, 16, 31, 32, 61, 44]. This issue even arises outside of universal approximation, e.g., various papers demonstrate the ability of RNNs and transformers to simulate various computational models such as Turing machines and automata, but require strong reliance on arbitrary precision [48, 42, 29, 9]. Infinite precision can inflate the expressivity of an architecture (function class) in a unrealistic and misleading way: for example, finite width RNNs with infinite precision can simulate Turing machines, but finite-precision, finite-width RNNs cannot. This is implied by streaming lower bounds [1] – any finite-precision, finite-width RNN induces a finite-space streaming algorithm corresponding to running

36th Conference on Neural Information Processing Systems (NeurIPS 2022).

the RNN on the inputs. However, streaming lower bounds tell us that finite-space streaming algorithms are not powerful enough to simulate Turing machines, and hence finite-precision, finite-width RNNs cannot either. As another example, Park et al. [41] exploit infinite precision in the parameters to show that a neural net with parameter count sublinear in $n$ can memorize $n$ arbitrary input-label pairs. However, a simple counting argument reveals that this result cannot be proven using finite precision networks – there are $2^n$ input-labeling pairs, but only $2^{o(n)}$ finite precision networks with $o(n)$ parameters.

More broadly, the ideal theoretical perspective should consider not only whether target functions can be expressed, but also whether the approximating functions can plausibly be obtained by fitting a neural network to a finite training sample, as is the case in practical deep learning settings. The latter question can be decomposed into studying optimization and generalization. Unfortunately, a rigorous analysis of optimization is unresolved even for simple two-layer nets [35, 33]. Global optimization analyses such as NTK do exist [18, 26], but there is a large body of theoretical and empirical work showing that neural networks can generalize much better than NTK analyses can hope to prove [20, 57]. Generalization is more tractable, so we propose to study expressivity and generalization together.

Towards studying more meaningful notions of approximation, this work proposes *statistically meaningful (SM) approximation*. This definition requires not only the existence of an approximating network, but also that it has good statistical properties. Consider a setting where the aim is to fit the target function $G$ using the approximating family $\mathcal{F}$ and a finite sample of training data. SM approximation requires existence of a loss whose empirical risk minimizer in $\mathcal{F}$ leads to a model with low approximation error in fitting $G$. We define the sample complexity of the approximation as the number of training samples needed to guarantee $\epsilon$ approximation error and study SM approximation with low sample complexity bounds. SM approximation essentially eliminates statistical concerns about fitting the target function with a finite sample (optimization concerns can remain).

We present two case studies on SM approximation. First, we demonstrate that overparameterized feedforward neural nets can SM approximate boolean circuits with a low sample complexity that depends only on the intrinsic circuit size. Though it is simple to construct neural nets to approximate boolean circuits, bounding the sample complexity of the approximation is challenging. For example, standard norm-based generalization bounds for the naive construction scale exponentially in depth [5, 6]. Furthermore, VC dimension-based bounds would scale polynomially in the number of parameters in the network [23], which is problematic because for practical optimization concerns, neural nets are typically overparameterized in terms of width [62]. In contrast, our sample complexity bound for SM approximation depends only on the intrinsic circuit size, up to logarithmic factors.

Our second case study is on SM approximating Turing machines with transformers. We consider a class of Turing machines with bounded computation time $T$ and construct encoder-decoder transformers [53] which SM approximate these Turing machines. The sample complexity of the approximation depends on a polynomial in $\log T$ and the sizes of the state space and the alphabet of the Turing machine. Though constructions for approximating Turing machines from prior work [48, 42, 9] have not been formally studied from a sample complexity perspective, existing bounds would depend at least linearly on $T$. Furthermore, our construction only uses $\log \log T$ precision, compared to at least $\log T$ in prior works, resulting in the exponential improvement in the sample complexity.

Proving sample complexity guarantees for our SM approximation results is nontrivial and requires additional insights. To obtain our sample complexity bounds, we leverage a recent generalization bound which depends on data-dependent Lipschitzness [56]. We develop theoretical tools to convert a broad class of neural nets, with possibly large Lipschitzness, into ones with small Lipschitzness on the training data, by introducing a number of new layers that is linear in depth. Our result applies to neural nets where each entry in the hidden representations on the training data takes values from a finite set (e.g., binary entries), and may be of independent interest.

In summary, our conceptual contribution is to propose a new notion of statistically meaningful approximation, intended to provide more meaningful guarantees by requiring that the approximating family have good statistical learnability. Technically, 1) we prove that feedforward neural nets can meaningfully approximate boolean circuits with sample complexity that depends polynomially on the width and depth of the circuit; and 2) we show that transformers can meaningfully approximate Turing machines with sample complexity logarithmic in the computation time.

## 1.1 Related works

Classical approximation theory for neural networks has a long history. Hornik et al. [25], Cybenko [16], and Leshno et al. [31] show that neural nets with one hidden layer are universal approximators but require the hidden layer size to grow exponentially in input dimension. Barron [4] uses the Fourier

transform to write target functions as infinite-width networks and subsamples neurons to obtain widths which depend only on target function properties. Lee et al. [30], Ji et al. [27] prove recent related developments in this direction of universal approximation.

Many works study benefits of deep networks over shallow ones [8, 2, 50, 19, 17, 11, 10]. Bengio and Delalleau [8] show separation for exact representation, whereas Telgarsky [50] shows separation for approximate representations with univariate inputs. Eldan and Shamir [19] demonstrate high-dimensional functions that can be approximated by two-layer polynomial-sized neural networks, but cannot be approximated by one-layer neural nets with subexponential hidden units. Via reduction to certain complexity theoretic questions, Vardi and Shamir [52] show that proving constant depth separations may be hard. Malach et al. [34] analyze the relationship between optimization and approximability, showing in various settings that deeper networks cannot be optimized if shallow networks cannot approximate them. This demonstrates that depth separation results [50] from approximation theory can be misleading since gradient descent anyways cannot optimize the deep networks used to construct the approximation.

Another area of study is on the ability of deep networks to memorize training data [62, 60, 41, 54]. Yun et al. [60] show that $\Theta(n)$ parameters are sufficient to memorize $\Theta(n)$ training points for ReLU nets with at least 3 layers, and Park et al. [41] reduce the parameter requirement to sublinear in $n$. Similar results have been proven for residual architectures [22] and convolutional nets [39]. Bartlett et al. [7] analyze the VC-dimension of neural nets, leading to bounds on the parameter count needed to fit training data. Other works study expressivity via connections to tensor approximation and sum-product networks [14, 15].

There is a long line of work on studying the ability of neural nets to recognize and represent formal languages. The seminal work of Siegelmann and Sontag [48] shows that RNNs are Turing complete but leverages infinite precision in the hidden activations. Chen et al. [12] extend this result to ReLU activations and study implications in language modeling. Many variants of transformers are shown to be Turing-complete, but these constructions also rely on arbitrary precision [42, 9]. Recent works have also proven results for generating or recognizing formal languages with finite-precision neural nets [58, 29, 24], but these results do not consider Turing machines or analyze statistical properties of their constructions. Concurrent work [13] proves Turing completeness of RNNs with finite precision, relying on a dynamically growing memory module in the architecture (which serves the same purpose as the long decoder sequences in our Transformer construction). However, they do not analyze statistical properties, which requires additional complications in both the construction and statistical analysis.

## 1.2 Notation

Let $f \circ g$ denote the composition of functions $f$ and $g$. For a family of functions $\mathcal{G}$, let $f \circ \mathcal{G} \triangleq \{f \circ g : g \in \mathcal{G}\}$ denote the family of compositions between $f$ and functions in $\mathcal{G}$. For a set $\mathcal{S}$ and function $f : \mathcal{S} \to \mathcal{Y}$, let $f(\mathcal{S})$ denote the set $\{f(s) : s \in \mathcal{S}\} \subseteq \mathcal{Y}$. We use $\mathbf{1}_d$ to denote the all-one's vector in $d$ dimensions, with the subscripted omitted if clear. For $i \in [d]$, we let $\mathbf{1}_d(i)$ denote the one-hot embedding in $d$-dimensions, which is 1 at index $i$ and 0 everywhere else. We use the notation $\widetilde{O}(\cdot)$ to hide poly-logarithmic factors in the argument. The notation $\lesssim, \gtrsim$ indicates the existence of a constant factor such that the inequality holds, and $\asymp$ denotes that the $\gtrsim$ and $\lesssim$ relations simultaneously hold. We use $\mathrm{poly}(\cdot)$ to indicate the existence of a polynomial in the argument which makes the equation true. For a set $\mathcal{A}$ (e.g., the set of alphabet symbols for a Turing machine) let $\mathcal{A}^*$ denote the set of all sequences of elements of $\mathcal{A}$, where sequence length can vary. Let $P$ denote a distribution over a space of inputs $\mathcal{X}$. Let $\xi_1, \ldots, \xi_n$ be $n$ i.i.d. Rademacher variables sampled from $\{-1, +1\}$. The expected $n$-sample Rademacher complexity of $\mathcal{F}$ on $P$ is as follows: $\mathrm{Rad}_{n,P}(\mathcal{F}) \triangleq \mathbb{E}_{(x_i)_{i=1}^n \overset{i.i.d}{\sim} P} \left[ \mathbb{E}_{\xi_1, \ldots, \xi_n} \left[ \sup_{F \in \mathcal{F}} \frac{1}{n} \sum_{i=1}^n \xi_i F(x_i) \right] \right]$, where $(x_i)_{i=1}^n$ denotes $n$ i.i.d. samples from $P$.

## 2 Statistically meaningful approximation

We consider settings where we wish to approximate every member $G$ in a real-valued function class $\mathcal{G}$ with some function $F$ in function class $\mathcal{F}$. Functions in both $\mathcal{G}$ and $\mathcal{F}$ map input space $\mathcal{X}$ to $\mathbb{R}$. In this work, $\mathcal{F}$ is some family of neural networks. Fix a loss $\ell : \mathbb{R} \times \mathbb{R} \to [0,1]$. The classical notion of $\epsilon$-approximation [43] is as follows:

**Definition 2.1** (Classical $\epsilon$-approximation)**.** *We say a function class $\mathcal{F}$ $\epsilon$-approximates a function class $\mathcal{G}$ with respect to loss $\ell$ and input distribution $P$, if for any given $G \in \mathcal{G}$, there exists $F \in \mathcal{F}$ such that $\mathbb{E}_{x \sim P}[\ell(F(x), G(x))] \leqslant \epsilon$.*

The issue with this classical notion of approximation is that it allows solutions which use infinite precision (or other potential unrealistic characteristics). Because of these drawbacks, even if $\mathcal{F}$ approximates $\mathcal{G}$, it does not mean that we can use $\mathcal{F}$ to fit the target function from $\mathcal{G}$ with a good sample complexity.

This work studies a stronger notion of approximation, statistically meaningful (SM) approximation, to eliminate statistical issues with fitting $G$ on a finite sample. SM-approximation requires that $\mathcal{G}$ is learnable via empirical risk minimization using models from $\mathcal{F}$, when data is generated from $P$.

**Definition 2.2** ($\epsilon$-SM-approximation). *We say a function class $\mathcal{F}$ $\epsilon$-SM-approximates a function class $\mathcal{G}$ with respect to evaluation loss $\ell$ and input distribution $P$ with sample complexity $n$ if there exists a surrogate loss $\bar{\ell} : \mathcal{F} \times \mathcal{X} \times \mathbb{R} \to [0,1]$ such that for any given $G \in \mathcal{G}$, the following holds:*

*With probability 0.99 over the randomness of $n$ examples $(x_i)_{i=1}^n$ drawn from $P$, the empirical risk minimizer of $\bar{\ell}$, $\widehat{F} \triangleq \arg\min_{F \in \mathcal{F}} \frac{1}{n} \sum_{i=1}^n \bar{\ell}(F, x_i, G(x_i))$, approximates $G$: $\mathbb{E}_{x \sim P}[\ell(\widehat{F}(x), G(x))] \leqslant \epsilon$.*

Definition 2.2 requires that the empirical risk minimizer of $\bar{\ell}$ over $\mathcal{F}$ on a finite sample $(x_i, G(x_i))_{i=1}^n$ is guaranteed to $\epsilon$-approximate $G$ on the population. Note that the surrogate loss $\bar{\ell}$ and evaluation loss $\ell$ can differ, and that $\bar{\ell}$ takes the model $F$ as an argument, allowing the empirical risk to include regularization.

Though Definition 2.2 may be reminiscent of PAC-learnability, there is a major conceptual difference: SM approximation unifies expressivity and generalization, whereas PAC-learnability is only concerned with generalization. For example, in the realizable PAC-learning case, there is no notion of an approximating family $\mathcal{F}$ – the setting only cares about fundamental learnability of $\mathcal{G}$. Furthermore, in agnostic PAC-learning (non-realizable) settings, the main focus is achieving a low loss *relative* to the best function in the hypothesis class. In contrast, SM approximation also requires *proving* that the best function in $\mathcal{F}$ achieves near-zero loss, whereas there is no such requirement in PAC-learning settings.

## 2.1 Background and tools

To prove SM-approximation guarantees, Definition 2.2 requires a loss surrogate $\bar{\ell}$ such that the empirical risk minimizer of $\bar{\ell}$ on the training data can approximate functions in $\mathcal{G}$. The following proposition, which is motivated by classical generalization theory, provides several conditions on $\bar{\ell}$ which lead to SM-approximation guarantees.

**Proposition 2.3.** *For loss function $\ell : \mathbb{R} \times \mathbb{R} \to [0,1]$ and input distribution $P$, suppose there exists a surrogate loss $\bar{\ell} : \mathcal{F} \times \mathcal{X} \times \mathbb{R} \to [0,1]$ satisfying the following properties:*

*1) For all $F \in \mathcal{F}$, $x \in \mathcal{X}$, $y \in \mathbb{R}$, $\bar{\ell}(F, x, y) \geqslant \ell(F(x), y)$.*

*2) For any $G \in \mathcal{G}$, consider the function class $\mathcal{L}_G \triangleq \{x \mapsto \bar{\ell}(F, x, G(x)) : F \in \mathcal{F}\}$. Then the $n$-sample Rademacher complexity of $\mathcal{L}_G$ is bounded: $\mathrm{Rad}_{n,P}(\mathcal{L}_G) \leqslant \epsilon$.*

*3) For any $G \in \mathcal{G}$, there exists $F \in \mathcal{F}$ with small surrogate loss: $\mathbb{E}_{x \sim P}[\bar{\ell}(F, x, G(x))] \leqslant \epsilon$.*

*Then, the function class $\mathcal{F}$ $O\left(\epsilon + \frac{1}{\sqrt{n}}\right)$-SM-approximates $\mathcal{G}$ with respect to loss $\ell$ and input distribution $P$ with sample complexity $n$.*

By Proposition 2.3, it suffices that $\bar{\ell}$ upper bounds the target loss $\ell$ and has low complexity, and $\mathcal{F}$ approximates $\mathcal{G}$ with respect to $(\bar{\ell}, P)$ in the classical sense. The proof follows from standard techniques for bounding generalization based on Rademacher complexity and is provided in Section A.

**All-layer margin loss.** We introduce one particular construction for $\bar{\ell}$ used in subsequent sections, which is motivated by the all-layer margin generalization bound proposed by [56]. This bound is based on data-dependent Lipschitzness measures [36, 55], and can provide stronger guarantees than classical norm-based bounds [37, 6, 38, 21].

We focus on the binary classification setting, where $G(x) \in \{0,1\}$, and study approximation with respect to the 0-1 loss $\ell_{0\text{-}1}(z, y) \triangleq \mathbb{1}((y - 0.5)z \leqslant 0)$ where $y \in \{0,1\}$ is assumed to be a binary label, and the aim is to output a negative prediction $z$ for $y = 0$ and positive for $y = 1$. We consider a family of functions $\mathcal{F}$ parameterized by $p$-dimensional parameters $\theta \in \Theta \subseteq \mathbb{R}^p$, such that $\mathcal{F} = \{x \mapsto F(x, \theta) : \theta \in \Theta\}$, where we abuse notation and let $F$ denote a general parameterized function $F : \mathcal{X} \times \mathbb{R}^p \to \mathbb{R}$. We sometimes use $\theta$ to identify an element of $\mathcal{F}$. Throughout the paper, we define $\Theta$ as a set with $\|\cdot\|_1$-norm bounded by $\alpha$:

$\|\theta\|_1 \leqslant \alpha$, $\forall \theta \in \Theta$. We define the parameter-based all-layer margin $\rho_F : \mathbb{R}^p \times \mathcal{X} \times \{0,1\} \to \mathbb{R}$ as follows:

$$\rho_F(\theta, x, y) \triangleq \min_{\delta} \|\delta\|_2$$
$$\text{subject to } (y - 0.5) \cdot F(x, \theta + \delta) \leqslant 0 \tag{2.1}$$

We omit $F$ from the subscript of $\rho$ when it is clear from context. This quantity measures the stability of the model around an input $x$ in parameter space. As is the case for the standard output margin, a larger all-layer margin, or better stability, tends to imply better generalization.

We modified the definition in [56] to consider perturbations $\delta$ in parameter space, whereas Wei and Ma [56] consider perturbations to the hidden layers. The parameter-space formulation is simpler and subsumes the results in [56]. Our formulation also accounts for weight sharing, which is important for our Turing machine results, whereas the formulation of [56] could not.

A key and immediate property of the all-layer margin is that it is strictly positive if and only if $F(x, \theta)$ predicts the correct label. We can leverage this property to construct a surrogate loss. For some parameter $\gamma$ intended to lowerbound the all-layer margins, we define the loss $\bar{\ell}_\gamma$ as follows:

$$\bar{\ell}_\gamma(\theta, x, y) = \begin{cases} 1 \text{ if } \rho(\theta, x, y) \leqslant 0 \\ 1 - \frac{\rho(\theta, x, y)}{\gamma} \text{ if } 0 < \rho(\theta, x, y) \leqslant \gamma \\ 0 \text{ if } \rho(\theta, x, y) \geqslant \gamma \end{cases} \tag{2.2}$$

Note that $\bar{\ell}_\gamma$ composes the classical ramp loss, which is used to prove margin-based generalization complexity bounds, with the value of the all-layer margin. By our construction, it immediately follows that $\bar{\ell}_\gamma(\theta, x, G(x)) \geqslant \ell_{0\text{-}1}(F(x, \theta), G(x))$, as is required of a surrogate loss.

We show that to obtain sample complexity bounds for SM-approximation of $\mathcal{G}$ in a classification setting, it suffices to prove that functions in $\mathcal{F}$ can fit labels of $G \in \mathcal{G}$ with large all-layer margin. Our argument uses $\bar{\ell}_\gamma$ as the loss surrogate in the definition of SM approximation. Though $\bar{\ell}_\gamma$ is computationally intractable to optimize, Wei and Ma [56] demonstrate that heuristically minimizing $\bar{\ell}_\gamma$ also leads to improved generalization empirically.

**Lemma 2.4.** *Fix any parameterized function $F : \mathcal{X} \times \mathbb{R}^p \to \mathbb{R}$, and define $\mathcal{F}_\alpha \triangleq \{x \mapsto F(x, \theta) : \theta \in \Theta\}$, where we assume $\Theta \subseteq \mathbb{R}^p$ is such that $\|\theta\|_1 \leqslant \alpha$ for all $\theta \in \Theta$. Fix $\epsilon \geqslant 0$. Suppose that for all $G \in \mathcal{G}$, there exists $\theta \in \Theta$ such that the following holds:*

$$\mathbb{E}_{x \sim P}\left[\mathbb{1}(\rho_F(\theta, x, G(x)) < \gamma)\right] \leqslant \epsilon \tag{2.3}$$

*Then, $\mathcal{F}_\alpha$ $\epsilon$-SM-approximates $\mathcal{G}$ with respect to $(\ell_{0\text{-}1}, P)$ with sample complexity $\widetilde{O}\left(\frac{1}{\epsilon^2}\left(\frac{\alpha^2 \log(p)}{\gamma^2} + 1\right)\right)$.*

Here $\widetilde{O}$ hides poly-logarithmic factors in the arguments, in this case, $\text{polylog}(\frac{\alpha^2 \log(p)}{\gamma^2 \epsilon^2})$ factors. The proof follows [56] and is deferred to Section A. In Section A, we also state a generalization bound for 0-1 loss based on (2.1), which may be of independent interest. We use (2.2) and Lemma 2.4 to prove that neural nets can SM-approximate Boolean circuits and Turing machines.

# 3  SM approximation of Boolean circuits with feedforward nets

This section shows that feedforward neural nets can SM-approximate Boolean circuits with sample complexity that depends polynomially on the size of the circuit. A boolean circuit $G : \{0,1\}^m \to \{0,1\}$ on $m$ inputs bits is described by a directed acyclic graph, with vertices of this graph referred to as "gates". The graph contains $m$ input gates of indegree 0, which are identified with the input bits. The remaining gates each compute a boolean function taking values at their parents as arguments, and a designated output gate produces the output of the entire circuit. We consider boolean circuits consisting of AND, OR, and NOT gates, which compute the corresponding boolean functions on 2, 2, and 1 inputs, respectively and are sufficient to compute any boolean function [45]. We also allow identity (ID) gates, which take 1 input and output the same value.

We consider layered circuits, where we can partition the gates into layers such that the only edges in the graph occur from gates in layer $i$ to gates in layer $i+1$ for some $i$. Note that we can transform any boolean circuit into a layered one by adding ID gates. Letting $q$ denote the number of layers and $r$ the maximum number of gates in any layer, we say that the circuit has depth $q$ and width $r$. We say

that a circuit with $s$ total gates has size $s$. Our convention will be that the set of input gates is considered a layer, so $r \geqslant m$. We consider the following class of boolean circuits:

$$\mathcal{G}_{q,r,s} = \{G : \{0,1\}^m \to \{0,1\} : G \text{ computed by circuit with depth } q, \text{ size } s, \text{ and width } r\}$$

We will approximate $\mathcal{G}_{q,r,s}$ using a family of width $w$, depth $d$ feedforward ReLU nets parameterized by linear weights and biases $\theta = (W_0, b_0, \dots, W_d, b_d)$ computed as follows: $F_{w,d}(x, \theta) = W_d \phi(W_{d-1} \phi(\cdots \phi(W_0 x + b_0) \cdots) + b_{d-1}) + b_d$, where all intermediate layers have width $w$ for simplicity and $\phi$ denotes the coordinate-wise ReLU activation. The weight parameters are set so that for $1 \leqslant i \leqslant d-1$, $W_i \in \mathbb{R}^{w \times w}$, $W_0 \in \mathbb{R}^{w \times m}$, and $W_d \in \mathbb{R}^{1 \times w}$. The bias parameters are such that $b_i \in \mathbb{R}^w$ for $0 \leqslant i \leqslant d-1$, and $b_d \in \mathbb{R}$. To control the sample complexity, we restrict our attention to parameters with total $\|\cdot\|_1$-norm bounded by $\alpha$, giving the following function class:

$$\mathcal{F}_{w,d,\alpha} = \{x \mapsto F_{w,d}(x,\theta) : \|\theta\|_1 \leqslant \alpha\}$$

The following theorem states that feedforward neural nets can statistically meaningfully approximate boolean circuits with sample complexity polynomial in the circuit size.

**Theorem 3.1.** *Consider the class $\mathcal{G}_{q,r,s}$ of size-$s$,width-$r$, and depth-$q$ layered boolean circuits, and the class $\mathcal{F}_{w,d,\alpha}$ of neural nets above. Suppose $w \gtrsim r$, $\alpha \asymp s$, and $d \asymp q$.*

*Then, for all $\epsilon > 0$ and any input distribution $P$ over $\{0,1\}^m$, $\mathcal{F}_{w,d,\alpha}$ $\epsilon$-SM-approximates $\mathcal{G}$ with respect to $(\ell_{0\text{-}1}, P)$ with sample complexity $\mathrm{poly}(s) \widetilde{O}\left(\frac{\log(wd)}{\epsilon^2}\right)$.*

We note that the bound in Theorem 3.1 only scales logarithmically in the width $w$ of the network, even if $w$ is arbitrarily greater than the circuit width $r$. This ensures that even heavily overparameterized nets will have low sample complexity of the approximation.

For this setting, the all-layer margin loss in (2.2) is essential for proving tight sample complexity bounds, as other surrogate losses $\bar{\ell}$ would give weaker results. For example, if we choose $\ell_{0\text{-}1}$ as the surrogate loss, VC-dimension bounds [23] imply that $\mathcal{F}_{w,d,\alpha}$ statistically meaningfully approximates $\mathcal{G}_{q,r,s}$ with sample complexity scaling in $\mathrm{poly}(wq)$ under the conditions of Theorem 3.1. This suffers a polynomial dependence on the overparameterized width $w$, which is not ideal for realistic settings, where neural nets are often wider than necessary to facilitate optimization. In contrast, our dependence on $w$ is logarithmic. Another possible surrogate loss is the *output* margin-based ramp loss, which can be used to prove norm-based sample complexities [6]. However, these bounds depend on $\prod_{i=1}^{d} \|W_i\|_{\mathrm{op}}$ (or related quantities), which would be exponentially large in $d$ for the naive construction in Section 3.1.

## 3.1 Proof sketch for Theorem 3.1

There are two key steps in the proof. First, given any layered circuit $G \in \mathcal{G}$, we construct a neural net that directly simulates $G$ by computing the layers of $G$ one-by-one, which is simple to do by directly constructing ReLU and linear layers to simulate the AND, OR, NOT, and ID gates.

**Lemma 3.2.** *In the setting of Theorem 3.1, let $G$ denote the layered boolean circuit, which we aim to compute using a neural net. Let $g_i : \{0,1\}^{r_{i-1}} \to \{0,1\}^{r_i}$ denote function computed between the $i{-}1$-th and $i$-th layers of $G$, which we assume have $r_{i-1}$ and $r_i$ gates, respectively, so $G = g_{q-1} \circ \cdots \circ g_1$.*

*Then there exist functions $f_1, \dots, f_{q-1}$, where each $f_i$ is computed by a feedforward ReLU net with two linear and activation layers, such that for all $i \in [q-1]$ and $x \in \{0,1\}^m$, $f_i \circ \cdots \circ f_1(x) = g_i \circ \cdots \circ g_1(x)$. Thus, the composition $F(\cdot, \theta) \triangleq f_{q-1} \circ \cdots \circ f_1$ satisfies $F(x, \theta) = G(x)$ for all $x \in \{0,1\}^m$. Note that we omitted the dependency of $f_{q-1}, \dots, f_1$ on parameters $\theta$ for simplicity.*

**Lower bounding all-layer margin.** The next step for proving SM-approximation is to construct a loss $\bar{\ell}$ so that the empirical risk minimizer of $\bar{\ell}$ on the training data has good sample complexity. This crucially requires the all-layer margin tool developed in Section 2.1, as other complexity measures (e.g. norm-based) would not give good sample complexity bounds.

Recall that the all-layer margin $\rho_F(\theta, x, G(x))$ measures the stability of the output $F(x, \theta)$ to perturbations in to $\theta$, and, by Lemma 2.4, it suffices to show that $F$ has large all-layer margin on $x \in \{0,1\}^m$. Unfortunately, we cannot guarantee that the naive construction from Lemma 3.2 has large all-layer margin without further modifications. To remedy this issue, Theorem D.6 introduces a generic way to convert the model $F(\cdot, \theta)$, with possibly small all-layer margin on $x \in \{0,1\}^m$, into a new architecture and parameter set $F'(\cdot, \theta')$, with provably large all-layer margin on $x \in \{0,1\}^m$, such that $F'(x, \theta') = F(x, \theta)$ on all inputs $x \in \{0,1\}^m$. The construction relies on introducing new layers to $F$ to obtain $F'$ and increases the total number of layers by only a constant factor. This step of the proof is formally stated in the following lemma.

**Lemma 3.3.** *In the setting of Lemma 3.2, let $F(\cdot,\theta) = f_{q-1} \circ \cdots \circ f_1$ be the neural net with parameters $\theta$ constructed to compute the circuit $G$. There exist "correction functions" $\zeta_1,\ldots,\zeta_{q-2}$, where $\zeta_i$ is computed by a neural net with two activation and linear layers, such that the composition $F'(\cdot,\theta') \triangleq f_{q-1} \circ \zeta_{q-2} \circ f_{q-2} \circ \cdots \circ \zeta_1 \circ f_1$ has large all-layer margin: $\rho_{F'}(\theta',x,G(x)) \geqslant \frac{1}{\text{poly}(s)}$ for all $x \in \{0,1\}^m$. Here $\theta'$ denotes the collection of all parameters, and dependency of $f_i,\zeta_i$ on $\theta'$ is omitted for simplicity.*

We convey the core intuitions for Lemma 3.3 in a simplified toy setting as follows. Consider the case where we start with an initial architecture $f$ computing $f(x,(W_1,\ldots,W_d)) = \left(\prod_{i=1}^{d} W_i\right)x - 0.5$, where $W_i \in \mathbb{R}$. In this simplified setting, we consider $W_i = 1 \; \forall i$. For input $x = 1$ and target $y = 1$, the all-layer margin is small: $\rho_f((1,\ldots,1),1,1) \lesssim \frac{1}{\sqrt{d}}$, where the architecture is in the subscript. Indeed, choosing $\delta_i = \frac{3}{d}$, we have $f(1,(1-\frac{3}{d},\ldots,1-\frac{3}{d})) = (1-\frac{3}{d})^d - 0.5 \approx \exp(-3) - 0.5 < 0$. Thus, by the definition of all-layer margin, $\rho_f((1,\ldots,1),1,1) \leqslant \sqrt{\sum_i \delta_i^2} \lesssim \frac{1}{\sqrt{d}}$.

Now we will insert ReLU layers in $f$ to increase the all-layer margin to $\Omega(1)$. We use ReLU layers to implement the round function, which has the key property that $\text{round}(z) = 1 \; \forall z \geqslant 2/3$.

**Proposition 3.4.** *For any $z \in \mathbb{R}$, we can implement the function* $\text{round}(z) = \begin{cases} 0 & \text{if } z < 1/3 \\ 3x - 1 & \text{if } 1/3 \leqslant z < 2/3 \\ 1 & \text{if } z \geqslant 2/3 \end{cases}$

*via a feedforward ReLU net, as follows:* $\text{round}(z) = 3\phi(z-1/3) - 3\phi(z-2/3)$.

We consider the following function $\widetilde{f}$, which inserts round between every layer in $f$:

$$\widetilde{f}(x,(W_1,\ldots,W_d)) = \text{round}(W_d\text{round}(W_{d-1}\cdots\text{round}(W_1 x)\cdots)) - 0.5 \tag{3.1}$$

For this demonstration, we ignore the parameters of round, though the actual proof considers them. The following claim shows that (3.1) preserves the output of $f$ while increasing the all-layer margin:

**Claim 3.5.** *In the setting above, $\widetilde{f}(1,(1,\ldots,1)) = f(1,(1,\ldots,1))$ and $\rho_{\widetilde{f}}((1,\ldots,1),1,1) \geqslant \frac{1}{3}$.*

This reflects a significant increase in the all-layer margin, while only increasing depth by a constant factor. The proof is simple: we observe that if $\delta_i \leqslant \frac{1}{3}$ for all $i$, the function output will not change because $\text{round}(z) = 1 \; \forall z \geqslant \frac{2}{3}$. This immediately gives the all-layer margin lower bound $\frac{1}{3}$.

To apply this construction more generally, we note that round corrects errors in previous layers. In the more general setting, we insert "correction functions" $\zeta$ between each layer satisfying the key property that $\zeta(h') = h$ if $h$ is the intended output of the layer and $h'$ is any perturbed value satisfying $\|h' - h\|_2 \leqslant \frac{1}{3}$. Since intended outputs of layers in the function constructed by Lemma 3.2 are binary-valued in $\{0,1\}^w$ because $F$ simulates a boolean circuit, we can simply apply the function round constructed in Proposition 3.4 elementwise as the correction function. By the construction, this can be implemented by adding two additional feedforward ReLU layers per correction function. Following the intuition for Claim 3.5, we prove that inserting these correction functions guarantees a large all-layer margin (Theorem D.6) on all $x \in \{0,1\}^m$. This leads to the proof of Lemma 3.3. We can complete the proof of Theorem 3.1 by invoking Lemma 2.4, as shown in Section B.

## 4   SM approximation of Turing machines with transformers

In this section, we show that transformers SM-approximate Turing machines with computation time bounded by $T$, using sample complexity polynomial in $\log(T)$ and the state space and alphabet sizes of the Turing machine. Constructions from prior work would require the approximation sample complexity to be linear in $T$ [48, 12, 42, 9]. Thus, we obtain an exponential improvement in the dependency on $T$.

We briefly describe a Turing machine; see [49] for a more thorough survey. A Turing machine is a model for computation specified by a tuple $(\mathcal{Z},\mathcal{A},S,\mathcal{Z}_{\text{term}})$ containing a set of states $\mathcal{Z}$, a tape alphabet $\mathcal{A}$, a transition function $S : \mathcal{Z} \times \mathcal{A} \to \mathcal{Z} \times \mathcal{A} \times \{-1,+1\}$, and set of terminal states $\mathcal{Z}_{\text{term}}$ indicating accept or reject. For simplicity, we assume the Turing machine has a single tape, as any single-tape Turing machine can simulate a multi-tape one with only quadratic increase in runtime [49]. Given an input $x \in \mathcal{A}^*$ recorded on the left-most part of the tape, the Turing machine performs computation in a sequence of timesteps. In each timestep, the machine determines the next state, symbol to write, and direction to move the head via the transition function.

We let $\mathrm{TM}_{(\mathcal{Z},\mathcal{A},S,\mathcal{Z}_{\text{term}})}$ denote the function computed by the Turing machine, which produces an output in $\{0,1\}$ (if the machine halts). Fixing the alphabet $\mathcal{A}$, we consider the class of binary functions computed by Turing machines with at most $k$ states terminating in $T$ steps:

$$\mathcal{G}_{k,T} \triangleq \{x \mapsto \mathrm{TM}_{(\mathcal{Z},\mathcal{A},S,\mathcal{Z}_{\text{term}})}(x) : |\mathcal{Z}| \leqslant k, \text{ and } \forall x \in \mathcal{X}, \mathrm{TM}_{(\mathcal{Z},\mathcal{A},S,\mathcal{Z}_{\text{term}})} \text{ terminates in } T \text{ steps} \} \quad (4.1)$$

Note that we can assume the input sequences $x$ also have length at most $T$, as this is the maximum computation time of the Turing machine and the maximum amount of symbols the Turing machine can read.

## 4.1 Transformer architecture for SM-approximating Turing machines

We study approximation of $\mathcal{G}$ with a family of architectures consisting of both an encoder and decoder component [53], described as follows. The encoder architecture is simple and only performs an embedding of the input symbols, using learnable symbol embeddings $E \in \mathbb{R}^{w \times |\mathcal{A}|}$ and fixed positional encodings $\beta(1), \beta(2), \ldots \in \mathbb{R}^w$. Given input $x \in \mathcal{A}^*$ with $m$ symbols, the encoder produces $m$ output vectors in $\mathbb{R}^w$ via $\mathrm{Enc}_i(x, E) = E_{:,x_i} + \beta(i)$, where $\mathrm{Enc}_i$ denotes the output of the encoder at the $i$-th position.

The decoder iteratively computes an output, running for $T$ steps. We define a transformer layer of the decoder as a sequence of modules consisting of decoder self-attention, followed by encoder-decoder attention, followed by three feedforward ReLU layers.

**Attention layers.** Attention layers consist of key, value, and query functions $K, V, Q$, each, computing a linear transformation. We omit parameters here for simplicity. For a single decoder timestep, the attention layer takes two types of inputs: a sequence of previously-computed representations $h_1, \ldots, h_i$, and a current input representation $h'$. The layer applies the key, value, and query functions as follows:

$$\tau_0, \tau_1, \ldots, \tau_i = Q(h')^\top K_0, Q(h')^\top K(h_1), \ldots, Q(h')^\top K(h_i)$$
$$v_0, v_1, \ldots, v_i = V_0, V(h_1), \ldots, V(h_i)$$

where $K_0$ and $V_0$ are fixed "null" key and value vectors which are learned parameters of the layer. Letting $\mathcal{J}$ denote the set of indices $\{j : \tau_j = \max\{\tau_0, \ldots, \tau_i\}\}$, the attention layer performs hard-max attention [42] to compute the output, as follows: $\mathrm{Attn}(h', (h_1, \ldots, h_i)) = h' + \frac{1}{|\mathcal{J}|} \sum_{j \in \mathcal{J}} v_j$.

Our theory also applies to the standard softmax attention used in practice, but we focus on the hard-max case for a simpler proof. Let $h_t^{(j)}$ denote the representation computed by the $j$-th layer of the decoder at timestep $t$. At timestep $i$, decoder self-attention at the $(j+1)$-th layer computes $\mathrm{Attn}(h_i^{(j)}, (h_1^{(j)}, \ldots, h_i^{(j)}))$. Letting $e_1, \ldots, e_m$ denote the encoder outputs, encoder-decoder self-attention at the $(j+1)$-th layer and $i$-th step would compute $\mathrm{Attn}(h_i^{(j)}, (e_1, \ldots, e_m))$.

**Transformer layers.** We use feedforward layers which apply 3 standard ReLU layers, as follows: $\mathrm{FF}(h) = \phi(W_3 \phi(W_2 \phi(W_1 h + b_1) + b_2) + b_3)$. Our theory also allows for residual feedforward layers, and the architecture here is chosen mainly to simplify the construction.

A transformer layer applies these constructions in sequence. Letting $H_i^{(j)} = (h_1^{(j)}, \ldots, h_i^{(j)})$ denote the output after the $j$-th transformer layer for timesteps $1 \leqslant t \leqslant i$, and $\theta^{(j)}$ the parameters, we compute

$$h_i^{(j+1,\text{dec})} = \mathrm{Attn}(h_i^{(j)}, H_i^{(j)}, \theta^{(j+1,\text{dec-attn})})$$
$$h_i^{(j+1,\text{enc})} = \mathrm{Attn}(h_i^{(j+1,\text{dec})}, (e_1, \ldots, e_m), \theta^{(j+1,\text{enc-attn})})$$
$$\mathrm{Tr}(h_i^{(j)}, H_i^{(j)}, (e_1, \ldots, e_m), \theta^{(j+1)}) = \mathrm{FF}(h^{(j+1,\text{enc})}, \theta^{(j+1,\text{ff})})$$

Note that we included the explicit dependence of the attention layers on the parameters for completeness. We now set $h_i^{(j+1)} = \mathrm{Tr}(h_i^{(j)}, H_i^{(j)}, (e_1, \ldots, e_m), \theta^{(j+1)})$.

**Decoder outputs.** We consider $d$-layer decoders, so $o_i \triangleq h_i^{(d)}$ denotes the output of the decoder at time $i$, which is also inputted to the decoder at time $i+1$ as follows: $h_{i+1}^{(0)} = h_i^{(d)} + \beta(i+1)$. The initial decoder input $h_0^{(0)}$ is a trainable parameter. The decoder runs for a fixed number of timesteps $T'$ and outputs prediction $\theta_{\text{cls}}^\top h_{T'}^{(d)}$. For simplicity, we assume $T' = T$, the computation time of the Turing machine family.

Note that our architecture allows long (length $T$) decoding sequences, whereas typical architectures in practice use decoding sequences with roughly the same length as the input [53]. The architecture we study is similar to ones studied by [42, 9].

We use $x \mapsto F_{w,d,T}(x, \theta)$ to denote the described transformer architecture with parameters $\theta$, $w$-dimensional hidden layers, $d$ transformer layers in the decoder, and $T$ decoder steps. This leads to the following class of transformer functions: $\mathcal{F}_{w,d,\alpha,T} = \{x \mapsto F_{w,d,T}(x,\theta) : \|\theta\|_1 \leqslant \alpha\}$. The following theorem states that this class of transformers SM-approximates the Turing machine family $\mathcal{G}$ defined in (4.1) with sample complexity polynomial in $\log T$, $k$ and $|\mathcal{A}|$.

**Theorem 4.1.** *In the setting above, consider the class $\mathcal{G}$ of functions computed by Turing machines with at most $k$ states, alphabet $\mathcal{A}$, and computation time bounded by $T$ steps for inputs $x \in \mathcal{X}$. Suppose that $w \gtrsim k|\mathcal{A}| + \log T$, $d \asymp \log T$, and $\alpha = \mathrm{poly}(k, |\mathcal{A}|, \log T)$.*

*Then, for all $\epsilon > 0$ and any input distribution $P$ over $\mathcal{X}$, $\mathcal{F}_{w,d,\alpha,T}$ $\epsilon$-SM-approximates $\mathcal{G}$ with respect to $(\ell_{0\text{-}1}, P)$ with sample complexity $\mathrm{poly}(k, |\mathcal{A}|, \log T) \widetilde{O}\left(\frac{\log(wd)}{\epsilon^2}\right)$.*

As with Section 3, we set the surrogate loss $\bar{\ell}$ in Definition 2.2 to be the all-layer margin loss defined in Section 2.1. Commonly-used alternatives for the surrogate loss would not suffice for either our construction or ones in prior work [48, 12, 42, 9]. First, the VC dimension of $\mathcal{F}_{w,d,\alpha,T}$ is at least $\Omega(wT)$. This is because transformer architectures which contain a decoder component can express RNNs, which by lower bounds have VC dimension at least $wT$ [28]. This indicates that using $\ell_{0\text{-}1}$ as the surrogate loss would lead to sample complexities that are suboptimal in both the overparameterized width $w$ and the computation $T$. Second, the correct norm-based Rademacher complexity bound to use for transformers is unclear; however, the RNN-based equivalent would scale with the $T$-th power of some parameter norm, or exponentially in $T$. Thus, as in Section 3, the all-layer margin surrogate loss (2.2) is essential for obtaining our sample complexity bounds.

## 4.2 Proof sketch for Theorem 4.1

Following Lemma 2.4, our goal is to construct a transformer which can simulate Turing machines with large all-layer margin, namely, $\Omega\left(\frac{1}{\mathrm{poly}(k, |\mathcal{A}|, \log T)}\right)$. The fundamental limitation of prior work [42] towards attaining this is that the positional embeddings are required to store values as small as $\frac{1}{\mathrm{poly}(T)}$. Our construction cannot afford to rely on values this small – informally, if the construction relies on the exact values of these small entries, then the all layer margin would be at most $\frac{1}{\mathrm{poly}(T)}$ because perturbing the layer by the small entries could change the prediction. Instead, we propose using $\mathrm{Bin}(i)$, the binary encoding of $i$ in $\lceil \log T \rceil$ bits, as the positional encoding for timestep $i$. This allows us to use unique positional encodings for each timestep which do not rely on arbitrary precision.

We describe the construction. Fix a Turing machine $G \in \mathcal{G}$. We first require notation to describe the computation of $G$. For input $x \in \mathcal{X}$, let $z_i(x)$, $a_i(x)$ denote the Turing machine state and symbol under the tape head at the end of step $i$. We let $l_i(x)$ denote the location of the Turing machine head at the conclusion of step $i$. During the timestep, the Turing machine computes $S(z_{i-1}(x), a_{i-1}(x))$, writes a new symbol under the head at location $l_{i-1}(x)$, and moves the head either left or right. Let $u_i(x)$ denote the symbol written during timestep $i$, and $q_i(x) \in \{\text{left}, \text{right}\}$ the movement direction of the head.

Following [42] with several key modifications, we simulate the Turing machine using the transformer as follows. Each timestep will maintain the invariance that $o_i$ contains an encoding of $z_i(x), a_i(x)$, and $l_i(x)$. Given that this invariance holds until timestep $i$, the transformer simulates timestep $i+1$ of the Turing machine with the following steps:

1) Use feedforward layers to apply transition $S$ on $z_i(x)$ and $a_i(x)$, which can be read from $o_i$, to obtain $z_{i+1}(x)$, $u_{i+1}(x)$, and movement direction $q_{i+1}(x) \in \{\text{left, right}\}$.

2) Using feedforward layers, compute $l_{i+1}(x)$ from $q_{i+1}(x)$ and the encoding of $l_i(x)$ in $o_i$.

3) Compute $a_{i+1}(x)$. We use decoder self-attention to search over past timesteps which wrote to $l_{i+1}(x)$. Our aim is to find $u_{i'}(x)$, where $i' = \max\{j \leqslant i+1 : l_{j-1}(x) = l_{i+1}(x)\}$. We implement a binary search over past timesteps $j$, which is needed to find the *largest* $j \leqslant i+1$ where $l_{j-1}(x) = l_{i+1}(x)$. The binary search is performed over the bits of $i'$ and can be implemented with $O(\lceil \log T \rceil)$ decoder self-attention layers, and the construction ensures large all-layer margin.

4) If no such $i'$ from the previous timestep existed, we check whether $l_{i+1}(x)$ contained an input symbol using encoder-decoder attention and copy this input symbol if so.

5) If no symbols were found in 3) or 4), $l_{i+1}(x)$ must contain the blank symbol (meaning it wasn't visited yet by the head). Thus, we have computed $a_{i+1}(x)$, so we have all the information needed to compute the new embedding $o_{i+1}$.

To lower bound the all-layer margin of the constructed transformer, we use Theorem D.6, which requires existence of a "correction function" which can correct outputs in previous layers. Since we construct a network with intermediate layer entries in $\{0,1\}$, we can use the same correction function as Section 3.1, which rounds to the nearest bit. The full proof is provided in Section C.

## 5 Conclusion

This work proposes a new definition of approximation, statistically meaningful approximation, which ensures that the approximating family not only has sufficient expressivity, but also exhibits good statistical learnability. Towards a first analysis with this definition, we show approximability of two function classes: boolean circuits and Turing machines, with strong sample complexity guarantees depending only on the intrinsic properties of these function classes. There are several interesting directions to extend our study of statistically meaningful approximation. Examples include proving more upper and lower bounds for statistically meaningful approximation for different target functions and neural net architectures, and using our definition as a lens to compare architectures.

### Limitations

One potential limitation is that the "correction function" machinery discussed in Lemma 3.3 relies on the discrete nature of boolean circuits and Turing machines, and so additional work and insight would be required to prove analogous SM-approximation results for continuous functions. One important property of discrete functions, which we suspect may be leveraged more generally, is that it is easy to correct errors in intermediate computations of discrete functions (by rounding). It would be interesting to see whether this property has a continuous analog which can be analyzed.

### Acknowledgements

CW was supported by a NSF Graduate Research Fellowship. YC is supported by Stanford Graduate Fellowship and NSF IIS 2045685. TM acknowledges support of Google Faculty Award, NSF IIS 2045685, and JD.com.

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
