# A  Proofs for Section 2

We prove Proposition 2.3 and Lemma 2.4.

*Proof of Proposition 2.3.* Let $(x_i)_{i=1}^n$ denote a $n$ i.i.d. training examples drawn from $P$ and fix $G \in \mathcal{G}$. Define $L(F) \triangleq \mathbb{E}_{x \sim P}[\bar{\ell}(F, x, G(x))]$ and $\widehat{L}(F) \triangleq \frac{1}{n}\sum_{i=1}^n \bar{\ell}(F, x_i, G(x_i))$. Let $\widehat{F} \in \mathcal{F}$ denote $\operatorname{argmin}_{F \in \mathcal{F}} \widehat{L}(F)$, the empirical risk minimizer of $\widehat{L}$, which we aim to show has population loss for fitting $G$ bounded by $O(\epsilon + \frac{1}{\sqrt{n}})$. By standard arguments using Rademacher complexity, we have with probability $1 - \delta$,

$$\sup_{F \in \mathcal{F}} |L(F) - \widehat{L}(F)| \leqslant 2\mathrm{Rad}_{n,P}(\mathcal{L}_G) + \sqrt{\frac{\log(2/\delta)}{n}}$$

$$\leqslant 2\epsilon + \sqrt{\frac{\log(2/\delta)}{n}} \tag{A.1}$$

Now note that by the condition 3) on $\bar{\ell}$, there exists $F^\star$ with $L(F^\star) \leqslant \epsilon$. Now we have

$$L(\widehat{F}) - L(F^\star) \leqslant (L(\widehat{F}) - \widehat{L}(\widehat{F})) + (\widehat{L}(\widehat{F}) - \widehat{L}(F^\star)) + (\widehat{L}(F^\star) - L(F^\star))$$

We bound the first and last term in parenthesis by applying (A.1), and the middle term is bounded by 0, by definition of $\widehat{F}$. It follows that

$$L(\widehat{F}) - L(F^\star) \leqslant 4\epsilon + 2\sqrt{\frac{\log(2/\delta)}{n}}$$

$$\implies L(\widehat{F}) \leqslant 5\epsilon + 2\sqrt{\frac{\log(2/\delta)}{n}}$$

where we used $L(F^\star) \leqslant \epsilon$. Finally, we use the fact that $\bar{\ell}$ upper bounds $\ell$, so $\mathbb{E}_{x \sim P}[\ell(\widehat{F}(x), G(x))] \leqslant L(\widehat{F})$. Plugging in $\delta = 0.01$ gives the desired result. $\qquad\square$

*Proof of Lemma 2.4.* We first observe that $\bar{\ell}_\gamma(\theta, x, y) \leqslant \mathbb{1}(\rho(\theta, x, y) < \gamma)$ by definition, so by (2.3), for all $G \in \mathcal{G}$ we have

$$\inf_{\theta \in \Theta} \mathbb{E}_{x \sim P}[\bar{\ell}(\theta, x, G(x))] \leqslant \epsilon$$

Thus, it remains to check the Rademacher complexity condition for applying Proposition 2.3. Fixing any $G \in \mathcal{G}$, define the function class $\mathcal{L}_G$ as in Definition 2.2.

We first observe that following the same argument as Claim A.4 of [56] (except we apply the perturbations to the parameters, rather than the hidden layers), $|\rho(\theta, x, y) - \rho(\theta', x, y)| \leqslant \|\theta - \theta'\|_2$ for any $\theta, \theta' \in \mathbb{R}^p$. Let $\mathcal{N}_{\|\cdot\|_2}(\varepsilon, \Theta)$ denote the $\varepsilon$-covering number of $\Theta$ in $\|\cdot\|_2$-norm, and $\mathcal{N}_{\|\cdot\|_\infty}(\varepsilon, \mathcal{L}_G)$ the $\varepsilon$-covering number of $\mathcal{L}_G$ in the norm defined by $\|H - H'\|_\infty = \max_{x \in \mathcal{X}} |H(x) - H'(x)|$ for any $H, H' \in \mathcal{L}_G$. The arguments of [56] imply that $\log \mathcal{N}_{\|\cdot\|_\infty}(\varepsilon, \mathcal{L}_G) \leqslant \log \mathcal{N}_{\|\cdot\|_2}(\gamma\varepsilon, \Theta) \leqslant O\left(\left\lceil \frac{\alpha^2 \log(p)}{\gamma^2 \varepsilon^2} \right\rceil\right)$, where the last inequality is from standard covering number bounds for $\|\cdot\|_1$ balls. Now we can apply this covering number bound in the Dudley entropy integral, another standard step to bound Rademacher complexity, to obtain that for all $n$, $\mathrm{Rad}_{n,P}(\mathcal{L}_G) \lesssim \frac{\alpha \log n \sqrt{\log(p)}}{\gamma \sqrt{n}}$ (see arguments in [56] for more detail). Solving for $n$ such that the r.h.s. of this equation is bounded by $\epsilon$ gives the desired result. $\quad\square$

Note that from the proof of Lemma 2.4, we would also obtain the following parameter-space all-layer margin generalization bound as a corollary, which may be of independent interest:

**Corollary A.1.** *In the setting of Lemma 2.4, let $Q$ denote a distribution over $(x, y)$ pairs, with $(x_i, y_i)_{i=1}^n$ denoting a set of $n$ i.i.d. training samples from $Q$. With probability $1 - \delta$ over the draw of the training samples, all classifiers $F(\cdot, \theta) \in \mathcal{F}$ which achieve zero 0-1 training loss satisfy*

$$\mathbb{E}_{x \sim Q}[\ell_{0\text{-}1}(F(x, \theta), y)] \leqslant O\left(\frac{\alpha\sqrt{\log(p)}}{\sqrt{n}}\sqrt{\frac{1}{n}\sum_{i=1}^n \frac{1}{\rho(\theta, x_i, y_i)^2}}\right) + \xi \tag{A.2}$$

*where $\xi \lesssim O\left(\frac{\log(1/\delta) + \log(n)}{\sqrt{n}}\right)$ is a low-order term.*

The proof of Corollary A.1 simply follows by plugging in the covering number bound on $\rho$ derived in the proof of Lemma 2.4 into Lemma 2.2 of [56].

# B  Proofs for Section 3

This section completes the proof of Section 3. The following lemma formally states that we can construct the neural net to simulate the circuit layerwise.

**Lemma B.1.** *In the setting of Theorem 3.1, let $G$ denote the layered boolean circuit, which we aim to compute using a neural net. Let $G_i : \{0,1\}^{r_{i-1}} \to \{0,1\}^{r_i}$ denote function computed between the $i-1$-th and $i$-th layers of $G$, which we assume have $r_{i-1}$ and $r_i$ gates, respectively. Let $f$ denote the following 2-layer neural net architecture, parameterized by $\theta = (W_1, b_1, W_2, b_2)$:*

$$f(h, \theta) = \phi(W_2\phi(W_1 h + b_1) + b_2)$$

*Then there exist $\theta$ with $\|\theta\|_1 = O(r_i)$ such that for any $h \in \{0,1\}^{r_{i-1}}$,*

$$f(\widetilde{h}, \theta) = \widetilde{G_i(h)}$$

*where $\widetilde{h}$ takes $h$ and appends $w - r_{i-1}$ zeros, and likewise for $\widetilde{G_i}(h)$.*

We note that the proof of Lemma 3.2 follows by applying Lemma B.1 $q-1$ times. Using Lemma B.1, we can complete the proof of Theorem 3.1.

*Proof of Theorem 3.1.* Our proof will construct a neural network to compute any boolean circuit with all-layer margin lower bound $\frac{1}{\text{poly}(r,q)}$. By Lemma 2.4, this will be sufficient to guarantee meaningful approximation.

There are two steps in our construction: first, given any layered circuit $G \in \mathcal{G}_{q,r,s}$, we construct a neural net that directly simulates $G$ by computing the layers of $G$ one-by-one. Our construction shows that we can compute every layer in $G$ using two feedforward ReLU layers, and results in a neural net $\widehat{F}$ computing $G$, but with possibly small all-layer margin. The next step is to convert $\widehat{F}$ into a neural net with large all-layer margin, i.e., implement Lemma 3.3. To do this, we insert "correction functions" (Definition D.1) between every group of layers in $\widehat{F}$. These correction layers leverage the knowledge that unperturbed outputs of these layers should be contained in $\{0,1\}^w$ and perform elementwise rounding to map perturbed values back to $\{0,1\}^w$. Theorem D.6 formally shows that by introducing these correction layers can guarantee a lower bound on the all-layer margin roughly depending on the Lipschitz constants of each individual layer. Furthermore, each correction layer can be computed via two feedforward ReLU layers, so introducing the correction layers only increases depth by a constant factor.

We implement the proof plan by first applying Lemma B.1 $q$ times in order to obtain the function $\widehat{F}$ computing $G$ (with padding) mentioned above. The total $\|\cdot\|_1$-norm of the parameters so far is at most $s$. Now we use the correction function described in Proposition 3.4, which we apply coordinate-wise on non-padding coordinates. We apply the correction functions after each layer constructed in Lemma B.1. Note that each correction function requires at most double the width of the corresponding layer in the circuit, and the parameters for all correction functions add total $\|\cdot\|_1$-norm at most $O(s)$.

Note that at this point, minor modifications are still required in order to apply Theorem D.6. The neural net output is in $\{0,1\}^w$, not $\{-1,1\}$; we can remedy this by setting the last layer to compute the linear transformation $z \mapsto 2z - 1$ on the single non-padding coordinate corresponding to the output. Second, to make the depth of the architecture consistently $d$, we can add sequences of identity functions before this last linear layer just constructed, followed by correction layers, until each of the constructed approximating functions reaches the desired fixed depth $d$. This finally gives us parameters $\theta$ with $\|\cdot\|_1$-norm bound $O(s + d)$, so that the set of constructed functions is contained in $\mathcal{F}_{w,d,\alpha}$. Thus, we showed that for $G \in \mathcal{G}_{q,r,s}$, there exists $\theta$ such that $F(x, \theta) = 2G(x) - 1$ for all $x \in \{0,1\}^m$.

Finally, it is straightforward to check that Condition D.3 for Theorem D.6 is satisfied for Lipschitzness parameters which are polynomial in the circuit width $r$. Thus, we apply Theorem D.6 to obtain a lower bound $\widehat{\gamma} = \frac{1}{\text{poly}(r,q)} \geqslant \frac{1}{\text{poly}(s)}$ on the all-layer margin for every input $x \in \{0,1\}^m$. Finally, we directly apply Lemma 2.4 using $\gamma = \widehat{\gamma}$ to obtain the desired result. $\qquad\square$

The following proposition will be used to construct basic gates in the circuit with a simple feedforward ReLU network.

**Proposition B.2.** *Let* $x = \begin{bmatrix} x_1 \\ x_2 \end{bmatrix} \in \{0,1\}^2$ *be binary inputs to* AND *and* OR *gates. The following feedforward ReLU networks compute the* AND *and* OR *functions:* $F_{\text{AND}}(x) = \phi(x_1 + x_2 - 1)$*, and* $F_{\text{OR}}(x) = 1 - \phi(1 - x_1 - x_2)$*.*

*Proof of Lemma B.1.* Each row of $W_1$ and value in $b_1$ will correspond to a single entry in the output of $\widetilde{G}_i$. The same applies for $W_2, b_2$. $W_2$ will be set to a diagonal matrix with entries in $\{-1, 0, 1\}$. For the 0 entries which only serve to pad the dimension, we set corresponding values in $W_1, b_1, W_2, b_2$ to be 0. For the remainder of the entries of $\widetilde{G}_i$ corresponding to actual gates in the circuit, in the case that the gates compute AND or OR, we fill in the values of corresponding rows in $W_1, b_1, W_2, b_2$ to implement the constructions for AND and OR in Proposition B.2. The construction for ID and NOT are even simpler. For example, to implement $\text{NOT}(z) = 1 - z$ for $z \in \{0, 1\}$ on coordinate $j$, we can set the $j$-th row of $W_1$ to have -1 on the diagonal and 0 everywhere else, $(b_1)_j = 1$, $(b_2)_j = 0$, and $(W_2)_{j,j} = 1$. It is easy to check that $\|\theta\|_1 = O(r_i)$ with this construction. $\qquad\square$

## C   Proof of Theorem 4.1

### C.1   Additional setup and notation

We fix any Turing machine $G \in \mathcal{G}$ and construct a transformer which can simulate $G$. Throughout this section, a superscript will be used to index layer indices, and a subscript to index timesteps.

We assume that the initial state of the tape has the input written at the left-most positions. The Turing machine always starts at a fixed initial state $z_{\text{init}}$. We let $[\varnothing] \in \mathcal{A}$ denote the blank symbol, which initially fills all positions on the tape which aren't part of the input. We construct a transformer that simulates the Turing machine up until it reaches a terminal state in $\mathcal{Z}_{\text{term}}$, at which the transformer will loop in that state until it hits a computation time $T$.

We introduce some notation which will appear throughout the construction. Define $w_{\text{pos}} \triangleq \lceil \log_2 T \rceil$. We use $w_{\text{pos}}$ to denote the effective dimension of the position embedding, as only $w_{\text{pos}}$ coordinates will be non-zero. For $0 \leqslant i \leqslant T$, define $\text{Bin}(i) \in \mathbb{R}^{w_{\text{pos}}}$ to be the vector containing the binary encoding of $i$: $\text{Bin}(i)_j = 1$ if the binary representation of $i$ contains 1 in the $j$-th bit and 0 otherwise.

For simplicity, the proof will focus on the setting without overparameterization, where we choose the dimension $w = w_{\text{TM}} \triangleq |\mathcal{Z}| + 2|\mathcal{A}| + 3w_{\text{pos}} + w_{\text{scr}}$ for storing all the hidden representations of the model, where $w_{\text{scr}} = O(w_{\text{pos}} + |\mathcal{A}| + |\mathcal{Z}|)$. We can extend our analysis to allow for arbitrary over-parameterization using $w > w_{\text{TM}}$ by designating a certain subset of the coordinates to always equal 0, and performing calculations using only a subset of $w_{\text{TM}}$ coordinates. We group the $w_{\text{TM}}$ coordinates using the following symbols: st for encoding the state, $\text{sym}_1$, $\text{sym}_2$ for encoding symbols, $\text{pos}_1$ and $\text{pos}_2$, $\text{pos}_3$ for encoding position, and scr, which is used as scratch space. Thus, for $h \in \mathbb{R}^w$, we can index its coordinates via the groups as follows:

$$
h = \begin{bmatrix}
h^{\text{st}} & \in \mathbb{R}^{|\mathcal{Z}|} \\
h^{\text{sym}_1} & \in \mathbb{R}^{|\mathcal{A}|} \\
h^{\text{sym}_2} & \in \mathbb{R}^{|\mathcal{A}|} \\
h^{\text{pos}_1} & \in \mathbb{R}^{w_{\text{pos}}} \\
h^{\text{pos}_2} & \in \mathbb{R}^{w_{\text{pos}}} \\
h^{\text{pos}_3} & \in \mathbb{R}^{w_{\text{pos}}} \\
h^{\text{scr}} & \in \mathbb{R}^{w_{\text{scr}}}
\end{bmatrix}
$$

When the meaning is clear from context, we use the superscript to index coordinate groups as described.

The position embedding $\beta(i)$ is defined formally so that $\beta(i)^{\text{pos}_1} = \text{Bin}(i)$, and $\beta(i)$ is 0 in all other coordinates. The encoder embedding matrix $E$ is such that

$$
\begin{aligned}
\text{Enc}_i(x)^{\text{sym}_1} &= \mathbb{1}_{|\mathcal{A}|}(x) \\
\text{Enc}_i(x)^{\text{pos}_1} &= \text{Bin}(i)
\end{aligned}
\tag{C.1}
$$

where $\text{Enc}_i(x)$ has 0's at all other coordinates. embedding function $e : \mathcal{A} \to \mathbb{R}^d$ for the encoder is defined such that $e(x)^{\text{sym}_1} = \mathbf{1}_{|\mathcal{A}|}(x)$, the one-hot encoding for $x \in \mathcal{A}$, and 0 everywhere else. We use $o_1, \ldots, o_T$ to refer to the output embeddings of the decoder. Our construction maintains the invariant

that the output embedding $o_i$ encodes $z_i(x)$, $a_i(x)$, $l_i(x)$ for each $i$. To achieve this, we maintain

$$o_i^{\text{st}} = \mathbf{1}_{|\mathcal{Z}|}(z_i(x))$$
$$o_i^{\text{sym}_1} = \mathbf{1}_{|\mathcal{A}|}(a_i(x)) \tag{C.2}$$
$$o_i^{\text{pos}_2} = \text{Bin}(l_i(x))$$

and $o_i$ has 0 at all other coordinates. Thus, the input $o_i + \beta(i+1)$ to the decoder at step $i+1$ is of the form

$$(o_i + \beta(i+1))^{\text{st}} = \mathbf{1}_{|\mathcal{Z}|}(z_i(x))$$
$$(o_i + \beta(i+1))^{\text{sym}_1} = \mathbf{1}_{|\mathcal{A}|}(a_i(x))$$
$$(o_i + \beta(i+1))^{\text{pos}_1} = \text{Bin}(i) \tag{C.3}$$
$$(o_i + \beta(i+1))^{\text{pos}_2} = \text{Bin}(l_i(x))$$

## C.2   Completing the proof

We implement the first step 1) in Section 4.2 using the following lemma. Note that the lemma uses two consecutive feedforward ReLU layers, but in our actual proof we will simulate this using two transformer layers where the attention parameters are all $\mathbf{0}$, and only the feedforward layers are instantiated.

**Lemma C.1.** *Let $\mathcal{O}$ denote the set of decoder inputs in the form* (C.3) *encoding $z_{i-1}(x)$, $a_{i-1}(x)$, $l_{i-1}(x)$ for some timestep $i$. For parameters $\theta = (W_1, b_1, W_2, b_2)$, consider the following function computing a sequence of two feedforward ReLU layers: $f(h,\theta) = \phi(W_2\phi(W_1 h + b_1) + b_2)$. There exist parameters $\theta$ such that for decoder inputs $h \in \mathcal{O}$,*

$$f(h,\theta)^{\text{st}} = \mathbf{1}_{|\mathcal{Z}|}(z_i(x))$$
$$f(h,\theta)^{\text{sym}_2} = \mathbf{1}_{|\mathcal{A}|}(u_i(x))$$
$$f(h,\theta)^{\text{pos}_1} = \text{Bin}(i) \tag{C.4}$$
$$f(h,\theta)^{\text{pos}_2} = \text{Bin}(l_{i-1}(x))$$

*Furthermore, $f(h,\theta)^{\text{scr}}$ will contain a one-hot encoding for $q_i(x)$, and besides this, $f(h,\theta)$ is 0 at all other coordinates. The parameters satisfy $\|\theta\|_1 = O(|\mathcal{Z}||\mathcal{A}| + w_{\text{pos}})$.*

*Proof.* We follow the construction used in Lemma B.2 of [42]. The first layer computes a one-hot encoding of the state, symbol input pair. We choose $W_1 : \mathbb{R}^{w_{\text{TM}}} \to \mathbb{R}^{|\mathcal{Z}||\mathcal{A}| + w_{\text{TM}}}$ so that the first $|\mathcal{Z}||\mathcal{A}|$ rows are described by:

$$(W_1)_{(z,a),:}^{\text{st}} = \mathbf{1}_{|\mathcal{Z}|}(z)$$
$$(W_1)_{(z,a),:}^{\text{sym}_1} = \mathbf{1}_{|\mathcal{A}|}(a)$$

and 0 everywhere else. The remaining rows of $w_{\text{TM}}$ rows of $W_1$ simply implement the identity mapping. We choose $b_1$ so that its first $|\mathcal{Z}||\mathcal{A}|$ entries are -1, and all other entries are 0. We observe that from this construction, for all $h \in \mathcal{O}$ where $h$ encodes $z_{i-1}(x), a_{i-1}(x)$,

$$\phi(W_1 h + b_1) = \begin{bmatrix} \mathbf{1}_{|\mathcal{Z}||\mathcal{A}|}((z_{i-1}(x), a_{i-1}(x))) \\ h \end{bmatrix}$$

This is because before the ReLU, the first $|\mathcal{Z}||\mathcal{A}|$ entries of $W_1 h$ will have 2 on the $(z_{i-1}(x), a_{i-1}(x))$-th entry and be bounded by 1 everywhere else, so adding $\alpha_1$ and applying the activation will zero out all but one entry.

Now it is simple to pick $W_2$ so that $f(h,\theta)$ is as desired because we can construct it to exactly encode the output of $S(z,a)$ for each of its first $(z,a)$ columns and copy over the other necessary entries of $h$ as needed by (C.4). $\square$

The next lemma demonstrates that we can use an additional sequence of feedforward ReLU layers to produce $\text{Bin}(l_i(x))$, given $\text{Bin}(l_{i-1}(x))$ and $q_i(x)$.

**Lemma C.2.** *In the setting of Theorem 4.1 and Lemma C.1 above, there is a function $f$ parameterized by $\theta$ composed of $O(w_{\text{pos}})$ feedforward ReLU layers such that for any $h$ computed by the function*

*in Lemma C.1 in the form* (C.4) *at timestep* $i$,

$$f(h,\theta)^{\text{st}} = \mathbf{1}_{|\mathcal{Z}|}(z_i(x))$$
$$f(h,\theta)^{\text{sym}_2} = \mathbf{1}_{|\mathcal{A}|}(u_i(x))$$
$$f(h,\theta)^{\text{pos}_1} = \text{Bin}(i) \tag{C.5}$$
$$f(h,\theta)^{\text{pos}_2} = \text{Bin}(l_{i-1}(x))$$
$$f(h,\theta)^{\text{pos}_3} = \text{Bin}(l_i(x))$$

*At all other coordinates,* $F(h, \theta)$ *takes value 0. Furthermore, the parameters satisfy* $\|\theta\|_1 = O(w_{\text{pos}}(|\mathcal{Z}| + |\mathcal{A}| + w_{\text{pos}}))$.

*Proof.* As the construction of Lemma C.1 encoded $q_i(x)$, the movement direction of the head, we can use feedforward ReLU layers to implement binary addition to either add or subtract 1 from $l_{i-1}(x)$. Let $v_1, v_2$ denote the bits in the scratch dimensions indicating the head movement, where $v_1 = 1, v_2 = 0$ indicates left and $v_1 = 0, v_2 = 1$ indicates right. Then more specifically, we first use $O(w_{\text{pos}})$ feedforward ReLU layers to compute $l_{i-1}(x) - v_1$, and then $O(w_{\text{pos}})$ additional feedforward ReLU layers to compute $l_{i-1}(x) - v_1 + v_2$. Note that the output would always be $l_i(x)$ by the definition of $v_1, v_2$.

It remains to implement a module which computes $\text{Bin}(j - v_1)$ given $v_1, \text{Bin}(j)$, and $\text{Bin}(j + v_2)$ given $v_2, \text{Bin}(j)$ for any $j \in [T]$. We can express the binary addition by a depth-$O(w_{\text{pos}})$ binary circuit, which can in turn be expressed by a neural net with $O(w_{\text{pos}})$ layers where each weight matrix has $\|\cdot\|_1$-norm $(|\mathcal{Z}| + |\mathcal{A}| + w_{\text{pos}})$ (which is required to implement the identity mapping to copy forward the other dimensions of $h$ which aren't involved in the binary addition). This gives the desired total $\|\cdot\|_1$-norm bound. $\square$

The next lemmas implement steps 3), 4), 5) in Section 4.2. For the following lemmas, it will be helpful to further index the scratch dimensions as follows: for a vector $h \in w_{\text{scr}}$,

$$h^{\text{scr}} = \begin{bmatrix} h^{\text{scr}_1} \in \mathbb{R}^{|\mathcal{A}|} \\ h^{\text{scr}_2} \in \mathbb{R}^{|\mathcal{A}|} \\ h^{\text{scr}_3} \in \mathbb{R}^{w_{\text{pos}}} \\ h^{\text{scr}_4} \in \mathbb{R}^3 \end{bmatrix}$$

**Lemma C.3.** *In the setting of Theorem 4.1 and Lemma C.2 above, fix any timestep $i$ and define* $i' = \max\{1 \leqslant t \leqslant i : l_{t-1}(x) = l_i(x)\}$. *If $j$ such that $l_{t-1}(x) = l_i(x)$ exists, we define $i' = 0$ otherwise. Consider any $H_i = (h_1, ..., h_i)$, where $h_t$ is computed by the layer in Lemma C.2 for timestep $t$, and in the form* (C.5). *There is a function $f$ parameterized by $\theta$ consisting of $O(w_{\text{pos}})$ total self-attention and linear layers such that for all such $H_i$, the following holds:*

$$f(h_i, H_i, \theta)^{\text{st}} = \mathbf{1}_{|\mathcal{Z}|}(z_i(x))$$
$$f(h_i, H_i, \theta)^{\text{sym}_2} = \mathbf{1}_{|\mathcal{A}|}(u_i(x))$$
$$f(h_i, H_i, \theta)^{\text{pos}_1} = \text{Bin}(i)$$
$$f(h_i, H_i, \theta)^{\text{pos}_2} = \text{Bin}(l_{i-1}(x))$$
$$f(h_i, H_i, \theta)^{\text{pos}_3} = \text{Bin}(l_i(x)) \tag{C.6}$$
$$f(h_i, H_i, \theta)^{\text{scr}_1} = \begin{cases} \mathbf{1}_{|\mathcal{A}|}(u_{i'}(x)) & \text{if } i' > 0 \\ \mathbf{0} & \text{otherwise} \end{cases}$$
$$F(h_i, H_i, \theta)_1^{\text{scr}_4} = \mathbb{1}(i' > 0)$$

*At all other coordinates,* $F(H, \theta)$ *takes value 0. Furthermore, the parameters satisfy* $\|\theta\|_1 = O(w_{\text{pos}}(|\mathcal{Z}| + |\mathcal{A}| + w_{\text{pos}}))$.

The proof plan will roughly implement a binary search to find $i'$, leveraging the attention layers. The first step in the binary search is to verify whether $i' > 0$, described below.

**Claim C.4.** *In the setting of Lemma C.3, let $H_i = h_1, ..., h_i$ be the input representations for timesteps $1, ..., i$. Suppose that each $h_t$ for $1 \leqslant t \leqslant i$ satisfies the following:*

$$h_t^{\text{pos}_1} = \text{Bin}(t)$$
$$h_t^{\text{pos}_2} = \text{Bin}(l_{t-1}(x)) \tag{C.7}$$

Additionally, suppose that $h_i$ is of the form in (C.5). Then there is a function $f^{(0)}$ parameterized by $\theta$ such that

$$f^{(0)}(h_i, H_i, \theta)^{\mathrm{scr}_1} = \mathbf{0}$$
$$f^{(0)}(h_i, H_i, \theta)^{\mathrm{scr}_3} = \mathbf{0} \qquad \text{(C.8)}$$
$$f^{(0)}(h_i, H_i, \theta)^{\mathrm{scr}_4}_1 = \mathbb{1}(i' > 0)$$

The function $f^{(0)}$ can be computed by a single decoder self-attention layer with $\|\theta\|_1 = O(w_{\mathrm{pos}})$.

Next, we implement the binary search itself, using $w_{\mathrm{pos}}$ self-attention layers. Each step of the binary search reveals a single bit of $i'$, so the $j$-th attention layer will compute a representation storing the $j$ most significant bits of $i'$. We let $\mathrm{Bin}_j(l) \in \{0,1\}^{w_{\mathrm{pos}}}$ to denote the binary encoding of the $j$ most significant bits of $l$: $(\mathrm{Bin}_j(l))_{j'} = (\mathrm{Bin}(l))_{j'}$ for $1 \leqslant j' \leqslant j$, and $(\mathrm{Bin}_j(l))_{j'} = 0$ for $j' > j$. We also set $\mathrm{Bin}_0(l) = \mathbf{0}$. We use the superscript $(j)$ to indicate the $j$-th set of layers in the binary search. The following claim implements each step of the binary search rigorously.

**Claim C.5.** *In the setting above and of Lemma C.3, let $H_i^{(j)} = h_1^{(j)}, \ldots, h_i^{(j)}$ be the representations computed after the $j$-th group of layers for timesteps $1$ through $i$, for $0 \leqslant j \leqslant w_{\mathrm{pos}} - 1$. Suppose that each $h_t^{(j)}$ for $1 \leqslant t \leqslant i$ satisfies the following:*

$$h_t^{(j),\mathrm{pos}_1} = \mathrm{Bin}(t)$$
$$h_t^{(j),\mathrm{pos}_2} = \mathrm{Bin}(l_{t-1}(x)) \qquad \text{(C.9)}$$

*In addition, suppose that $h_i^{(j)}$ satisfies:*

$$h_i^{(j),\mathrm{scr}_1} = \mathbf{0}$$
$$h_i^{(j),\mathrm{scr}_3} = \begin{cases} \mathrm{Bin}_j(i') & \text{if } i' > 0 \\ \mathbf{0} & \text{otherwise} \end{cases} \qquad \text{(C.10)}$$
$$(h_i^{(j),\mathrm{scr}_4})_1 = \mathbb{1}(i' > 0)$$

*with all other coordinates matching the quantities prescribed in (C.5). Then there is a function $f^{(j+1)}$ parameterized by $\theta$ such that*

$$f^{(j+1)}(h_i^{(j)}, H_i^{(j)}, \theta)^{\mathrm{scr}_1} = \mathbf{0}$$
$$f^{(j+1)}(h_i^{(j)}, H_i^{(j)}, \theta)^{\mathrm{scr}_3} = \begin{cases} \mathrm{Bin}_{j+1}(i') & \text{if } i' > 0 \\ \mathbf{0} & \text{otherwise} \end{cases} \qquad \text{(C.11)}$$
$$f^{(j+1)}(h_i^{(j)}, H_i^{(j)}, \theta)^{\mathrm{scr}_4}_1 = \mathbb{1}(i' > 0)$$

*with all other coordinates matching those prescribed in (C.5). We note that $f^{(j+1)}$ consists of a single decoder self-attention layer followed by single feedforward ReLU layer, with $\|\theta\|_1 = O(|\mathcal{Z}| + |\mathcal{A}| + w_{\mathrm{pos}})$.*

At the end of the $w_{\mathrm{pos}}$-th application of the binary search, we would have found $\mathrm{Bin}(i')$ exactly. It remains to apply another attention layer which attends directly to timestep $i'$ and copies $u_{i'}(x)$.

**Claim C.6.** *In the setting above and of Lemma C.3, let $H_i = h_1, \ldots, h_i$ be the representations computed after the $w_{\mathrm{pos}}$-th group of layers constructed in Claim C.5 for timesteps $1$ through $i$. Suppose that each $h_t$ for $1 \leqslant t \leqslant i$ satisfies the following:*

$$h_t^{\mathrm{sym}_2} = \mathbb{1}_{|\mathcal{A}|}(u_t(x))$$
$$h_t^{\mathrm{pos}_1} = \mathrm{Bin}(t) \qquad \text{(C.12)}$$
$$h_t^{\mathrm{pos}_2} = \mathrm{Bin}(l_{t-1}(x))$$

*In addition, suppose that $h_i$ satisfies:*

$$h_i^{\mathrm{scr}_1} = 0$$
$$h_i^{\mathrm{scr}_3} = \begin{cases} \mathrm{Bin}(i') & \text{if } i' > 0 \\ \mathbf{0} & \text{otherwise} \end{cases} \qquad \text{(C.13)}$$
$$(h_i^{\mathrm{scr}_4})_1 = \mathbb{1}(i' > 0)$$

with all other coordinates matching the quantities prescribed in (C.5). Then there is a function $f^{(w_{\text{pos}}+1)}$ parameterized by $\theta$ such that $f^{(w_{\text{pos}}+1)}(h_i, H_i, \theta)$ computes the desired output in (C.6). Furthermore, $f^{(w_{\text{pos}}+1)}$ consists of a single decoder self-attention layer followed by a single feedforward ReLU layer, and $\|\theta\|_1 = O(|\mathcal{Z}| + |\mathcal{A}| + w_{\text{pos}})$.

Putting these together, we complete the proof of Lemma C.3.

*Proof of Lemma C.3.* For the purposes of this proof, we index the layers by a superscript to avoid confusion with indexing timesteps. We set $f^{(0)}$ to be the function defined in Claim C.4. We note that layers output by $f^{(0)}$ satisfy the condition of Claim C.5, so we can apply Claim C.5 inductively to obtain layers $f^{(1)}, \ldots, f^{(w_{\text{pos}})}$ where their applying their composition results in representations satisfying (C.12) and (C.13). Now we set $f^{(w_{\text{pos}}+1)}$ to be the function constructed in Claim C.5, which gives the desired output. Finally, we note that by summing the $\|\cdot\|_1$ bounds for the parameters constructed in each layer, we can finally obtain $\|\theta\|_1 = O(w_{\text{pos}}(|\mathcal{Z}| + |\mathcal{A}| + w_{\text{pos}}))$. □

We fill in the proofs of Claims C.4, C.5, and C.6 below.

*Proof of Claim C.4.* To construct the decoder self-attention, the query function will be of the form $Q(h) = W_Q h + b_Q$ and $K(h) = W_K h + b_K$, where $W_Q, W_K \in \mathbb{R}^{(w_{\text{pos}}+1) \times w}$ and $b_Q, b_K \in \mathbb{R}^{w_{\text{pos}}+1}$. We choose the parameters such that the following equations hold:

$$Q(h)_{1:w_{\text{pos}}} = 2h^{\text{pos}_3} - 1$$
$$Q(h)_{w_{\text{pos}}+1} = 1$$

and

$$K(h)_{1:w_{\text{pos}}} = 2h^{\text{pos}_2} - 1$$
$$K(h)_{w_{\text{pos}}+1} = 0$$

The value function $V(h)$ is such that $V(h)_1^{\text{scr}_4} = 1$, and $V(h)_\ell = 0$ on all other coordinates $\ell$, which can be implemented by a linear transformer. Finally, we set the null key $K_0$ and value $V_0$ such that $(K_0)_{w_{\text{pos}}+1} = w_{\text{pos}} - 1$, with 0 everywhere else, and $V_0 = \mathbf{0}$. Letting $\theta_{\text{attn}}$ denote the attention parameters, the layer is of the form

$$f^{(0)}(h_i, H_i, \theta) = \text{Attn}(h_i, H_i, \theta)$$

To see that $f^{(0)}$ satisfies (C.8), observe that if $i' > 0$, $Q(h_i)^\top K(h_{i'}) = w_{\text{pos}}$ by (C.7) and construction of $Q, K$. On the other hand, $Q(h_i)^\top K_0 = w_{\text{pos}} - 1$. Thus, $\text{argmax}_t Q(h_i)^\top K(h_t) \in [i]$, which implies that $f^{(0)}(h_i, H_i, \theta)_1^{\text{scr}_4} = 1$ by the construction of $V$. In the other case where $i' = 0$, we note that $Q(h_i)^\top K(h_t) \leqslant w_{\text{pos}} - 2$ for all $1 \leqslant t \leqslant i$, so the null position is attended to. By construction of $V_0$, this implies $f^{(0)}(h_i, H_i, \theta)_1^{\text{scr}_4} = 0$. As $V, V_0$ are 0 on all other coordinates, it follows that (C.8) holds. It's also easy to observe that the $\|\theta\|_1$ is as desired. □

*Proof of Claim C.5.* The first layer in $f^{(j+1)}$ computes decoder self-attention. The query function is of the form $Q(h) = W_Q h + b_Q$, and the key function is of the form $K(h) = W_K h + b_h$, where $W_Q, W_K \in \mathbb{R}^{(w_{\text{pos}}+j+2) \times w}$ and $b_Q, b_K \in \mathbb{R}^{(w_{\text{pos}}+j+2)}$. We choose the parameters so that the following equations hold:

$$Q(h)_{1:w_{\text{pos}}} = 2h^{\text{pos}_3} - 1$$
$$Q(h)_{w_{\text{pos}}+1:w_{\text{pos}}+j} = 2h_{1:j}^{\text{scr}_3} - 1$$
$$Q(h)_{w_{\text{pos}}+j+1} = 1$$
$$Q(h)_{w_{\text{pos}}+j+2} = 1$$

and

$$K(h)_{1:w_{\text{pos}}} = 2h^{\text{pos}_2} - 1$$
$$K(h)_{w_{\text{pos}}+1:w_{\text{pos}}+j+1} = 2h_{1:j+1}^{\text{pos}_1} - 1$$
$$K(h)_{w_{\text{pos}}+j+2} = 0$$

Both of these functions can be constructed via linear transformations of $h$, with $\|W_Q\|_1 + \|W_K\|_1 + \|b_Q\|_1 + \|b_K\|_1 = O(w_{\text{pos}})$. Now we construct the value function $V(h) = W_V h + b_V$ such that $V(h)_3^{\text{scr}_4} = 1$ and $V(h)_\ell = 0$ on all other coordinates, which is also easily implemented by a linear layer. For the attention, the last quantities to construct are the null key $K_0$ and value $V_0$. $K_0$ will satisfy $(K_0)_{w_{\text{pos}}+j+2} = w_{\text{pos}} + j$, with 0 everywhere else. $V_0$ will simply be 0 on all coordinates. Letting $\theta_{\text{attn}} = (W_Q, b_Q, W_K, b_K, W_V, b_V, K_0, V_0)$ denote the attention parameters, the first layer will now be in the form

$$f^{(j+1),1}(h_i^{(j)}, H_i^{(j)}, \theta_{\text{attn}}) = \text{Attn}(h_i^{(j)}, H_i^{(j)}, \theta_{\text{attn}})$$

where Attn uses the constructed key, value, and query functions. We claim that $f^{(j+1),1}(h_i^{(j)}, H_i^{(j)}, \theta_{\text{attn}})$ satisfies the following:

$$f^{(j+1),1}(h_i^{(j)}, H_i^{(j)}, \theta_{\text{attn}})_3^{\text{scr}_4} = \begin{cases} 1 & \text{if } i' > 0 \text{ and has } (j+1)\text{-th bit } 1 \\ 0 & \text{otherwise} \end{cases} \tag{C.14}$$

For all other coordinates $\ell$, $f^{(j+1),1}(h_i^{(j)}, H_i^{(j)}, \theta_{\text{attn}})_\ell = (h_i^{(j)})_\ell$. To see this, we first observe that $Q(h_i^{(j)})^\top K_0 = w_{\text{pos}} + j$. Next, we observe that $Q(h_i^{(j)})_{1:w_{\text{pos}}}$ produces the encoding of $l_i(x)$ using binary $\{-1, +1\}$ bits, and $K(h_t^{(j)})_{1:w_{\text{pos}}}$ produces the encoding of $l_{t-1}(x)$ using binary $\{-1, +1\}$ bits by (C.9). In addition, $Q(h_i^{(j)})_{w_{\text{pos}}+1:w_{\text{pos}}+j} = 2\text{Bin}_j(i') - 1$ if $i' > 0$ and all 0's otherwise, and $K(h_t^{(j)})_{w_{\text{pos}}+1:w_{\text{pos}}+j+1} = 2\text{Bin}_{j+1}(t) - 1$. Note that by our construction, the maximum possible value of $Q(h_i^{(j)})^\top K(h_t^{(j)})$ is $w_{\text{pos}} + j + 1$, and the next largest possible value is $w_{\text{pos}} + j - 1$. Now there are 3 cases:

**Case 1:** $i' = 0$. In this case, we note that $l_i(x)$ never matches $l_{t-1}(x)$ for $1 \leq t \leq i$. Thus, by construction of the first $w_{\text{pos}}$ coordinates of $Q$ and $K$, the largest possible value of $Q(h_i^{(j)})^\top K(h_t^{(j)})$ is $w_{\text{pos}} + j - 1$, so the attention will always only attend to the null position, so the layer adds $V_0 = \mathbf{0}$ to $h_i^{(j)}$, preserving its value. Note that $(h_i^{(j), \text{scr}_4})_3 = 0$ in this case, which matches the desired behavior.

**Case 2:** $i' > 0$, and has $(j + 1)$-th bit 0. In this case, we note that for all $t > i'$, $Q(h_i^{(j)})^\top K(h_t^{(j)}) \leq w_{\text{pos}} + j - 1$, because by definition such $t$ must satisfy $l_{t-1}(x) \neq l_i(x)$, so the first $w_{\text{pos}}$ coordinates contribute at most $w_{\text{pos}} - 2$ to the dot product. On the other hand, if $t \leq i'$, $t$ must have $(j + 1)$-th bit 0, so $K(h_t^{(j)})_{w_{\text{pos}}+j+1} = -1$. This doesn't match the $(w_{\text{pos}} + j + 1)$-th bit of the query, so $Q(h_i^{(j)})^\top K(h_t^{(j)}) \leq w_{\text{pos}} + j - 1$ again. Thus, in this case, the null position is attended to again. The same reasoning as Case 1 then applies.

**Case 3:** $i' > 0$ and has $(j+1)$-th bit 1. In this case, $\max_t Q(h_i^{(j)})^\top K(h_t^{(j)}) = w_{\text{pos}} + j + 1$: for example, $t = i'$ achieves this maximum by our construction. As a result, the null position is not attended to. All the values in the positions attended to satisfy $V(h_t^{(j)})_3^{\text{scr}_4} = 1$, which matches the $(j + 1)$-th bit of $i'$. Thus, (C.14) holds.

Finally, to complete the proof we simply append an additional feedforward ReLU layer which copies the value $f^{(j+1),1}(h_i^{(j)}, H_i^{(j)}, \theta_{\text{attn}})_3^{\text{scr}_4}$ to the output bit corresponding to the position indexed by $\cdot_{j+1}^{\text{scr}_3}$. This layer will also set the output bit corresponding to $\cdot_3^{\text{scr}_4}$ to 0. Note that these operations can be implemented with a linear layer, and applying a ReLU activation after won't change the output, which is in $\{0,1\}^w$. By (C.10), the constructed function will thus satisfy (C.11). It's also easy to observe that $\|\theta\|_1$ is as desired. $\qquad \square$

*Proof of Claim C.6.* The attention layer uses key and query functions which each compute linear transformations from $\mathbb{R}^w$ to $\mathbb{R}^{2w_{\text{pos}}+1}$. The value function is also linear. We choose parameters such that

$$Q(h)_{1:w_{\text{pos}}} = 2h^{\text{pos}_3} - 1$$
$$Q(h)_{w_{\text{pos}}+1:2w_{\text{pos}}} = 2h^{\text{scr}_3} - 1$$
$$Q(h)_{2w_{\text{pos}}+1} = 1$$

and

$$K(h)_{1:w_{\text{pos}}} = 2h^{\text{pos}_2} - 1$$
$$K(h)_{w_{\text{pos}}+1:2w_{\text{pos}}} = 2h^{\text{pos}_1} - 1$$
$$K(h)_{2w_{\text{pos}}+1} = 0$$

and

$$V(h)^{\mathrm{scr}_1} = h^{\mathrm{sym}_2}$$

Furthermore, we choose null keys and positions such that $(K_0)_{2w_{\mathrm{pos}}+1} = 2w_{\mathrm{pos}} - 1$, and $V_0 = \mathbf{0}$. To follow the attention layer, we construct a linear layer which simply zeros out coordinates indexed by $\cdot^{\mathrm{scr}_3}$ and preserves all other coordinates. Note that because all outputs are either 0 or 1, applying a ReLU activation won't change the result. To see that this construction computes (C.6), we observe that if $i' > 0$, $Q(h_i)^\top K(h_{i'}) = 2w_{\mathrm{pos}}$. Otherwise, if $i' = 0$, $Q(h_i)^\top K(h_t) \leqslant 2w_{\mathrm{pos}} - 2$ for all $1 \leqslant t \leqslant i$. On the other hand, it always hold that $Q(h_i)^\top K_0 = 2w_{\mathrm{pos}} - 1$. Thus, if $i' > 0$, the attention attends exactly to $i'$, so the value function satisfies $V(h_{i'}) = \mathbf{1}_{|\mathcal{A}|}(u_{i'}(x))$, which would produce the output in (C.6), as desired. On the other hand, if $i' = 0$, the attention attends to the null position, so the attention layer sets $f^{(w_{\mathrm{pos}}+1)}(h_i, H_i, \theta)^{\mathrm{scr}_1} = \mathbf{0}$. Thus, $f^{(w_{\mathrm{pos}}+1)}$ also produces the desired output in this case. It's also easy to observe that the $\|\theta\|_1$ is as desired. $\qquad\square$

The next step is to complete step 4) in Section 4.2 using encoder-decoder attention. The following lemma provides this construction.

**Lemma C.7.** *In the setting of Theorem 4.1 and Lemma C.3, consider any timestep $i$ and let $h$ denote an output of the function constructed in Lemma C.3, in the form* (C.6). *Let $e_1, \ldots, e_m$ denote the outputs of the encoder, in the form* (C.1). *There is a function $f$ with parameter $\theta$ consisting of a single encoder-decoder attention layer such that for all such $h$ in the form* (C.6), *the following holds:*

$$
\begin{aligned}
f(h,(e_1,\ldots,e_m),\theta)^{\mathrm{st}} &= \mathbf{1}_{|\mathcal{Z}|}(z_i(x)) \\
f(h,(e_1,\ldots,e_m),\theta)^{\mathrm{sym}_2} &= \mathbf{1}_{|\mathcal{A}|}(u_i(x)) \\
f(h,(e_1,\ldots,e_m),\theta)^{\mathrm{pos}_1} &= \mathrm{Bin}(i) \\
f(h,(e_1,\ldots,e_m),\theta)^{\mathrm{pos}_2} &= \mathrm{Bin}(l_{i-1}(x)) \\
f(h,(e_1,\ldots,e_m),\theta)^{\mathrm{pos}_3} &= \mathrm{Bin}(l_i(x)) \\
f(h,(e_1,\ldots,e_m),\theta)^{\mathrm{scr}_1} &= \begin{cases} \mathbf{1}_{|\mathcal{A}|}(u_{i'}(x)) & \textit{if } i' > 0 \\ 0 & \textit{otherwise} \end{cases} \qquad\qquad\text{(C.15)} \\
f(h,(e_1,\ldots,e_m),\theta)^{\mathrm{scr}_2} &= \begin{cases} \mathbf{1}_{|\mathcal{A}|}(x_{l_i(x)}) & \textit{if } l_i(x) \leqslant m \\ \mathbf{0} & \textit{otherwise} \end{cases} \\
f(h,(e_1,\ldots,e_m),\theta)^{\mathrm{scr}_4}_1 &= \mathbb{1}(i' > 0) \\
f(h,(e_1,\ldots,e_m),\theta)^{\mathrm{scr}_4}_2 &= \mathbb{1}(l_i(x) \leqslant m)
\end{aligned}
$$

*At all other coordinates, $f(h,(e_1,\ldots,e_m),\theta)$ takes value $0$. Furthermore, the parameters satisfy $\|\theta\|_1 = O(|\mathcal{A}| + w_{\mathrm{pos}})$.*

*Proof.* We choose the encoder-decoder attention layer so that the key, value, and query functions are linear transformations. The key and query functions map $\mathbb{R}^w$ to $\mathbb{R}^{w_{\mathrm{pos}}+1}$ and compute the following:

$$
\begin{aligned}
Q(h)_{1:w_{\mathrm{pos}}} &= 2h^{\mathrm{pos}_3} - 1 \\
Q(h)_{w_{\mathrm{pos}}+1} &= 1
\end{aligned}
$$

and

$$
\begin{aligned}
K(h)_{1:w_{\mathrm{pos}}} &= 2h^{\mathrm{pos}_1} - 1 \\
K(h)_{w_{\mathrm{pos}}+1} &= 0
\end{aligned}
$$

The value function computes

$$
\begin{aligned}
V(h)^{\mathrm{scr}_2} &= h^{\mathrm{sym}_1} \\
V(h)^{\mathrm{scr}_4}_2 &= 1
\end{aligned}
$$

with 0's in all other coordinates. The null key $K_0$ satisfies $(K_0)_{w_{\mathrm{pos}}+1} = w_{\mathrm{pos}} - 1$, with 0's in all other coordinates. The null value $V_0$ satisfies $V_0 = \mathbf{0}$. We set

$$f(h,(e_1,\ldots,e_m),\theta) = \mathrm{Attn}(h,(e_1,\ldots,e_m),\theta)$$

where Attn is the decoder-encoder attention using the key, value, and query described above. Now we observe that from this construction, if $h$ is in the form provided in (C.6), then $Q(h)_{1:w_{\mathrm{pos}}} = \mathrm{Bin}(l_i(x))$.

In addition, we have $K(e_j)_{1:w_{\mathrm{pos}}} = e_j^{\mathrm{pos}_1} = \mathrm{Bin}(j)$ for $1 \leqslant j \leqslant m$. Thus, by construction of $V, K_0, V_0$, if $l_i(x) \leqslant m$, the attention attends to position $l_i(x)$ in the embedding. The value function for this position satisfies $V(e_{l_i(x)})^{\mathrm{scr}_2} = e_{l_i(x)}^{\mathrm{sym}_1} = \mathbb{1}_{|\mathcal{A}|}(x_{l_i(x)})$. Thus, in this case $F(h, \theta)$ computes the desired output in (C.15). On the other hand, if $l_i(x) > m$, then the attention will attend to the null position, as $Q(h)^\top K_0 = w_{\mathrm{pos}} - 1$, and the largest possible score for all other positions is $w_{\mathrm{pos}} - 2$. In this case, (C.15) holds again. It is also easy to check that the desired bound on $\|\theta\|_1$ would hold. $\qquad\square$

Finally, we implement step 5) of the outline in Section 4.2 in the following lemma.

**Lemma C.8.** *In the setting of Theorem 4.1 and Lemma C.7, consider any timestep $i$ and any $h$ output by the function in Lemma C.7 taking the form in* (C.15). *Then there is a function $f$ with parameters $\theta$ consisting of a constant number of feedforward ReLU layers satisfying the following:*

$$f(h, \theta)^{\mathrm{st}} = \mathbf{1}_{|\mathcal{Z}|}(z_i(x))$$
$$f(h, \theta)^{\mathrm{sym}_1} = \mathbf{1}_{|\mathcal{A}|}(a_i(x)) \qquad \text{(C.16)}$$
$$f(h, \theta)^{\mathrm{pos}_2} = \mathrm{Bin}(l_i(x))$$

*At all other coordinates, $F(h, \theta)$ takes values $0$. Furthermore, the parameters satisfy $\|\theta\|_1 = O(|\mathcal{Z}| + |\mathcal{A}| + w_{\mathrm{pos}})$.*

*Proof.* It suffices to construct a sequence of layers which performs the following operations:

1) Compute the following vector $v \in \mathbb{R}^3$:

$$v = \begin{cases} \begin{bmatrix} 1 \\ 0 \\ 0 \end{bmatrix} & \text{if } h_1^{\mathrm{scr}_4} = 1 \\[20pt] \begin{bmatrix} 0 \\ h_2^{\mathrm{scr}_4} \\ 1 - h_2^{\mathrm{scr}_4} \end{bmatrix} & \text{if } h_1^{\mathrm{scr}_4} = 0 \end{cases}$$

Note that $v$ encodes the location of the symbol $a_i(x)$, as $a_i(x) = u_{i'}(x)$ if $i' > 0$, $a_i(x) = x_{l_i(x)}$ if $i' = 0$ and $l_i(x) \leqslant m$, and $a_i(x) = [\varnothing]$ otherwise. The vector $v$ is a one-hot vector indicating which of these three cases holds.

2) We can take $v_1$ and compute AND with all bits of $h^{\mathrm{scr}_1}$, which computes $\mathbf{1}_{|\mathcal{A}|}(u_{i'}(x)) = \mathbf{1}_{|\mathcal{A}|}(a_i(x))$ if $i' > 0$, and $\mathbf{0}$ otherwise.

3) We take $v_2$ and compute AND with all bits of $h^{\mathrm{scr}_2}$, which computes $\mathbf{1}_{|\mathcal{A}|}(x_{l_i(x)})$ if $v_2 = 1$, and $\mathbf{0}$ otherwise.

4) We take $v_3$ and compute AND with all bits of $\mathbf{1}_{|\mathcal{A}|}([\varnothing])$, which computes $\mathbf{1}_{|\mathcal{A}|}(a_i(x))$ if $v_3 = 1$.

5) We add the outputs of 2), 3), and 4) together, which gives $\mathbf{1}_{|\mathcal{A}|}(a_i(x))$. We copy this quantity into the output coordinates indexed by $\cdot^{\mathrm{sym}_1}$. Then we set coordinates not listed in (C.16) to 0, producing the desired output.

Each of these operations can be computed by a constant number of feedforward ReLU layers, with total parameter norm satisfying $\|\theta\|_1 = O(|\mathcal{Z}| + |\mathcal{A}| + w_{\mathrm{pos}})$. $\qquad\square$

*Proof of Theorem 4.1.* We construct a neural net to compute any Turing machine with all-layer margin lower bound $\frac{1}{\mathrm{poly}(k, |\mathcal{A}|, \log T)}$ and apply Lemma 2.4 to turn this into a statement about statistically meaningful approximation.

For our Turing machine construction, we follow the outline laid out in Section 4.2. Fix any $G \in \mathcal{G}$. As mentioned, we first consider the case where $w = w_{\mathrm{TM}}$ exactly, as overparameterization is easy to deal with by always designating some subset of extra coordinates to be 0. We construct a transformer $\widehat{F}$ to compute $G$. First, we note that Lemma C.1 constructs a layer to compute the functionality described in 1). Next, the layer in Lemma C.2 performs the functionality in 2). Likewise, Lemmas C.3, C.7, C.8 construct layers which perform 3), 4), and 5). Thus, by applying the layers constructed from these

lemmas in sequence, we obtain a transformer such that the output $o_T$ contains an onehot encoding for $z_T(x)$: $\mathbf{1}_{|\mathcal{Z}|}(z_T(x))$. We can now apply a linear weight vector $\theta_{\mathrm{cls}}$ on the output to obtain $\theta_{\mathrm{cls}}^\top o_T$, where $(\theta_{\mathrm{cls}})_z = 1$ for accept states $z \in \mathcal{Z}_{\mathrm{term}}$ and $(\theta_{\mathrm{cls}})_z = -1$ for reject states. For inputs $x \in \mathcal{X}$, by our construction this computes the desired $\mathrm{TM}(x)$. Next, following Theorem 3.1, we insert correction functions (Definition D.1) between every group of constructed layers, which can be implemented via two feedforward ReLU layers following Proposition 3.4. The parameters for all correction functions add total $\|\cdot\|_1$-norm at most $\mathrm{poly}(k, |\mathcal{A}|, \log T)$. Let $\widehat{F}(x, \widehat{\theta})$ denote the transformer constructed this way, with parameters $\widehat{\theta}$. Note that for all $x \in \mathcal{X}$, $\widehat{F}(x, \widehat{\theta}) = 2G(x) - 1$.

Next, there are several steps remaining to convert $\widehat{F}$ into the fixed architecture $F_{w,d,T}^{\mathrm{tr}}$. First, we need to convert the layers in $\widehat{F}$ into transformer layers. This is achievable because every single decoder self-attention or encoder-decoder attention layer or feedforward ReLU module can be converted into a transformer layer by setting the two unused modules in the transformer layer to implement the identity function. This only increases the $\|\cdot\|_1$-norm by $\mathrm{poly}(k, |\mathcal{A}|, \log T)$. Note that in particular, we can perform this conversion such that the correction functions form the last 2 feedforward ReLU layers in every transformer layer. The first 3 layers in the transformer layer correspond to ones constructed in the lemmas. Second, we need to expand the dimension to a consistent width $w$. This is achievable by padding each layer with coordinates designated to be 0, without affecting any of the $\|\cdot\|_1$-norm bounds on the parameters. Third, we need to expand the depth to a fixed depth $d$. We can achieve this by appending transformer layers which compute the identity function (and also include correction functions) as needed.

Now we aim to apply Theorem D.6 by viewing the transformer as a very deep network with depth $d = O(T \log T)$, by applying each of the steps in the transformer computation in sequence. Note that our construction for the transformer layers is such that we can view the self-attention, encoder-decoder attention, and single feedforward ReLU layer as a single function in the setting of Theorem D.6. The correction function corresponds to the last 2 feedforward ReLU layers in the transformer layer. (We observe that there are actually $m$ layers which depend on the input $x$, not a single layer $f_0$ as in the setting of Theorem D.6, but this is a minor difference where the same argument of Theorem D.6 still easily applies.) Note that this network uses layer-based weight sharing, which is handled by Theorem D.6. Furthermore, the depth of this network doesn't affect the all-layer margin because Theorem D.6 doesn't depend on the number of layers. We also observe that Condition D.4 holds for $\lambda = \mathrm{poly}(|\mathcal{Z}|, |\mathcal{A}|, \log T)$, because all of the intermediate layers are sparse binary vectors with at most $|\mathcal{Z}| + |\mathcal{A}| + \log T$ nonzero entries.

Finally, it remains to check that Condition D.3 can hold for all of the defined layers for parameters that are polynomial in $|\mathcal{Z}|, |\mathcal{A}|, \log T$. This is straightforward to check for transformer layers where the attention layers have parameters $\mathbf{0}$, as standard results on the Lipschitzness of a single ReLU network would apply. For layers where the functionality comes from the attention mechanism, we observe that for valid inputs $x \in \mathcal{X}$, the largest attention score is always greater than the second largest by a margin of 1. Furthermore, ties only occur when all of the value vectors for the attended positions are already the same. As a result, the positions attended to by the layer will not change unless we perturb the parameters and inputs by $\Omega(\mathrm{poly}^{-1}(|\mathcal{Z}|, |\mathcal{A}|, \log T))$. This reasoning can be used to conclude that Condition D.3 with Lipschitz constants $\mathrm{poly}(|\mathcal{Z}|, |\mathcal{A}|, \log T)$, and distance parameters $\Omega(\mathrm{poly}^{-1}(|\mathcal{Z}|, |\mathcal{A}|, \log T))$ holds. As a result, the all-layer margin bound from applying Theorem D.6 will also be $\Omega(\mathrm{poly}^{-1}(|\mathcal{Z}|, |\mathcal{A}|, \log T))$, as desired. Finally, applying Lemma 2.4 with $\gamma = \Omega(\mathrm{poly}^{-1}(|\mathcal{Z}|, |\mathcal{A}|, \log T))$ and using the fact that the parameter $\|\cdot\|_1$-norms are bounded by $\alpha$ gives the desired result. $\qquad \square$

# D   All-layer margin lower bounds via correction functions

We consider a generalized architecture for a $d$-layer network as follows. Let $f_0 : \mathcal{X} \times \Theta_0 \to \mathbb{R}^w$ map space of inputs $x \in \mathcal{X}$ and parameters $\theta \in \Theta_0$ to $w$-dimensional space. For simplicity we assume all intermediate layers have dimension $w$, and let $f_i : \mathbb{R}^w \times \Theta_i \to \mathbb{R}^w$ be the $i$-th function in the neural net for $d > i \geqslant 1$. We define $f_d$ to output values in $\mathbb{R}$. Let $\theta = (\theta_0, ..., \theta_d) \in \Theta$ denote the full vector of parameters. The $i$-th hidden layer $h_i$ computes the following value, defined recursively:

$$h_0(x, \theta) = f_0(x, \theta_0)$$
$$h_i(x, \theta) = f_i(h_0(x, \theta), ..., h_{i-1}(x, \theta), \theta_i)$$

The model computes output $h_d(x, \theta)$. We will assume the existence of "correction" functions $\zeta$ parameterized by $\xi = (\xi_0, ..., \xi_{d-1}) \in \Xi_0 \times \cdots \times \Xi_{d-1}$ which correct errors in the model output for inputs $\mathcal{X}$:

**Definition D.1** (Correction functions). *Let $F' : \mathcal{X} \to \mathbb{R}$ be a model defined by layer functions $f_0, ..., f_d$. Then $\zeta_0, ..., \zeta_{d-1} : \mathbb{R}^w \to \mathbb{R}^w$, $\xi$ is a set of correction functions and parameters for $F'$, $\theta$ with radius*

$\sigma_\zeta$ if for all $i \in [d-1], x \in \mathcal{X}$ and $\widehat{h} \in \mathbb{R}^{\mathcal{X}}$ satisfying $\|\widehat{h} - h_i(x,\theta)\|_2 \leqslant \sigma_\zeta$,

$$\zeta_i(\widehat{h}, \xi_i) = h_i(x, \theta)$$

*We now define the function output $F$ with correction layers recursively by*

$$
\begin{aligned}
g_0(x,\theta,\xi) &= f_0(x,\theta_0) \\
\widetilde{h}_i(x,\theta,\xi) &= \zeta_i(g_{i-1}(x,\theta,\xi),\xi_i) \; \forall 0 \leqslant i \leqslant d-1 \\
g_i(x,\theta,\xi) &= f_i(\widetilde{h}_0(x,\theta,\xi),...,\widetilde{h}_{i-1}(x,\theta,\xi),\theta_i,\xi_i) \; \forall 1 \leqslant i \leqslant d \\
F(x,\theta,\xi) &= g_d(x,\theta,\xi)
\end{aligned}
\tag{D.1}
$$

*We note that for all $x \in \mathcal{X}$, $F(x,\theta,\xi) = h_d(x,\theta)$.*

The key observation is that by adding correction layers to the model, we can transform a model with possibly small all-layer margin on the input data to one with large all-layer margin. We first need to characterize the Lipschitzness of the individual layers.

**Definition D.2.** *We say that a function $f(\cdot,\theta) : \mathcal{D} \to \mathcal{D}_{\mathrm{out}}$ is $(\kappa_\theta,\mu,\sigma_h,\sigma_\theta)$-nice on $\mathcal{H} \subseteq \mathcal{D}$ with respect to $\|\!|\cdot|\!\|$ if the following hold:*

$$
\begin{aligned}
\|f(h,\theta) - f(h,\widehat{\theta})\|_2 &\leqslant \kappa_\theta \|\theta - \widehat{\theta}\|_2 \max\{\|\!|h|\!\|,1\} && \forall \|\theta - \widehat{\theta}\| \leqslant \sigma_\theta, h \in \mathcal{H} \\
\|f(h,\widehat{\theta}) - f(\widehat{h},\widehat{\theta})\|_2 &\leqslant \mu \|\!|h - \widehat{h}|\!\| && \forall \|\!|h - \widehat{h}|\!\| \leqslant \sigma_h, \|\theta - \widehat{\theta}\| \leqslant \sigma_\theta, h \in \mathcal{H}
\end{aligned}
$$

We will focus on the following norm on tuples of inputs $(v_1,...,v_i)$, where $h_j \in \mathbb{R}^w$ for all $j \in [i]$:

$$\|\!|(v_1,...,v_i)|\!\| = \max_j \|v_j\|_2 \tag{D.2}$$

We analyze the function $F$ output by a model with correction layers satisfying the following assumptions:

**Condition D.3.** *There are constants $\kappa_\theta, \kappa_\xi, \mu, \sigma_h, \sigma_\theta, \sigma_\zeta$ such that the following hold.*

*For $i \geqslant 1$, suppose that $f_i$ is $(\kappa_\theta, \mu, \sigma_h, \sigma_\theta)$-nice at $\theta_i$ on $(h_0,...,h_{i-1})(\mathcal{X})$ with respect to $\|\!|\cdot|\!\|$.*

*In addition, suppose that $f_0$ satisfies $\|f_0(x,\theta) - f_0(x,\widehat{\theta})\|_2 \leqslant \mu_0 \|\theta - \widehat{\theta}\|_2$ for all $x \in \mathcal{X}, \theta \in \Theta_0$.*

*Furthermore, suppose that for all $i$, $\zeta_i$ satisfies $\|\zeta_i(h,\xi_i) - \zeta_i(h,\widehat{\xi})\|_2 \leqslant \kappa_\xi \max\{\|h\|_2,1\}\|\xi_i - \widehat{\xi}\|_2$ for all $\widehat{\xi}$ with $\|\xi_i - \widehat{\xi}\|_2 \leqslant \sigma_\xi$ and $h \in \mathbb{R}^w$.*

These conditions are all standard Lipschitzness-based conditions on the individual layer functions. Our lower bound for the all-layer margin will be expressed in terms of the constants here.

We will also need to assume a bound $\lambda$ on the norms of each of the layers computed by $h_i$.

**Condition D.4.** *The norms of the true layer values are bounded, that is, $\exists \lambda$ such that for all $0 \leqslant i \leqslant d$ and $x \in \mathcal{X}$,*

$$\max\{\|h_i(x,\theta)\|_2, 1\} \leqslant \lambda \tag{D.3}$$

We will also consider models with weight sharing, which allows our analysis to apply to architectures such as the transformer in Section 4.

**Definition D.5** (Layer-based weight sharing). *Let $\Theta' \subseteq \mathbb{R}^{w'}, \Theta_0 \subseteq \mathbb{R}^{w_0},...,\Theta_d \subseteq \mathbb{R}^{w_d}$ be some spaces of real-valued parameters. Suppose we wish to perform copying on parameters $\theta' \in \Theta'$ to produce parameters $\theta = (\theta_0,...\theta_d) \in \Theta = \Theta_0 \times \cdots \Theta_d$, where $\theta_i$ is the set of parameters given to layer function $f_i$. We say that a tuple of functions $\tau = (\tau_0,...,\tau_d) : \Theta' \to \Theta$ is a layer-based weight sharing scheme if each $\tau_i$ is of the form*

$$\tau_i(\theta') = (\theta'_{\pi_1},...,\theta'_{\pi_{b_i}}) \tag{D.4}$$

*where $\pi_1,...,\pi_{b_i}$ is a set of distinct indices taking values in $[w']$. Note that this ensures that parameters are not duplicated within a layer.*

We will now prove our main lower bound for the all-layer margin based on inserting correction functions at every layer.

**Theorem D.6.** *In the above setting, suppose that Conditions D.3 and D.4 hold for a function $F$ in the form given by* (D.1) *parametrized by $\theta$ with correction layers $\zeta_0,...\zeta_{d-1}$ parameterized by $\xi$ with correction radius $\sigma_\zeta < 1$. Suppose that $F(x) \in \{-1,+1\}\; \forall x \in \mathcal{X}$. Then for all $x \in \mathcal{X}$, we can bound the all-layer margin of $F$ (defined in* (2.1))*as follows:*

$$\rho_F((\theta,\xi),x,\mathbb{1}(F(x,\theta,\xi)\geqslant 0)) \geqslant \min\{\frac{\lambda}{\mu_0},\frac{\sigma_\zeta}{\mu_0},\sigma_\theta,\sigma_\xi,\frac{1}{2\kappa_\theta},\frac{\sigma_\zeta}{2\kappa_\theta\lambda},\frac{\sigma_h}{2\kappa_\xi\lambda},\frac{\sigma_\zeta}{4\lambda\mu\kappa_\xi},\frac{1}{4\mu\kappa_\xi}\} \qquad (\text{D.5})$$

*Here the subscript $F$ makes it explicit that the all-layer margin is for the architecture $F$. Furthermore, if we consider any layer-based weight-shared model $F'(x,\theta') \triangleq F(x,\tau^{(1)}(\theta'),\tau^{(2)}(\theta'))$ for valid weight-tying mappings $\tau^{(1)}$, $\tau^{(2)}$ (Definition D.5), the same bound holds for $\rho_{F'}(\theta',x,\mathbb{1}(F'(x,\theta')\geqslant 0))$.*

Our proof will first consider the case without weight sharing. We use $\widehat{\theta} = (\widehat{\theta}_0, \dots, \widehat{\theta}_d)$ and $\widehat{\xi} = (\widehat{\xi}_0,...,\widehat{\xi}_{d-1})$ to denote a perturbed set of parameter vectors. Furthermore, define the partially perturbed parameter sets $\widehat{\theta}_i \triangleq (\widehat{\theta}_0,...,\widehat{\theta}_i,\theta_{i+1},...,\theta_d)$ and $\widehat{\xi}_i \triangleq (\widehat{\xi}_0,...,\widehat{\xi}_i,\xi_{i+1},...,\xi_d)$. We also use $\widehat{\theta}_{-1} \triangleq \theta$ and $\widehat{\xi}_{-1} \triangleq \xi$ when convenient.

We consider perturbations such that the following norm bounds hold:

$$\|\widehat{\theta}_0 - \theta_0\|_2 \leqslant \min\{\frac{\lambda}{\mu_0},\frac{\sigma_\zeta}{\mu_0}\} \qquad (\text{D.6})$$

$$\|\widehat{\theta}_i - \theta_i\|_2 \leqslant \min\{\sigma_\theta,\frac{1}{2\kappa_\theta},\frac{\sigma_\zeta}{2\kappa_\theta\lambda}\} \qquad (\text{D.7})$$

$$\|\widehat{\xi}_i - \widehat{\xi}_i\|_2 \leqslant \min\{\sigma_\xi,\frac{\sigma_h}{2\kappa_\xi\lambda},\frac{\sigma_\zeta}{4\lambda\mu\kappa_\xi},\frac{1}{4\mu\kappa_\xi}\} \qquad (\text{D.8})$$

We show that such perturbations won't change the label predicted by the model, and so therefore the minimum of these quantities immediately gives a lower bound on the all-layer margin. Our proof will be by induction, with the following lemma providing the base case.

**Lemma D.7.** *In the setting of Theorem D.6, suppose that* (D.6) *holds. Then the following hold:*

$$\widetilde{h}_0(x,\widehat{\theta},\xi) = h_0(x,\theta)$$

$$\|g_0(x,\widehat{\theta},\widehat{\xi}) - h_0(x,\theta)\|_2 \leqslant \min\{\lambda,\sigma_\zeta\}$$

The next lemma provides the inductive step. Starting with the base case, we show that because of the presence of the correction functions, the perturbations with our given bounds won't change the next layer output by too much. This allows the correction function to fix the output of the next layer, and this argument can extend inductively.

**Lemma D.8.** *In the setting of Theorem D.6, fix some $1 \leqslant i \leqslant d$. Suppose that for all $0 \leqslant j < i$, it holds that for all $x \in \mathcal{X}$,*

$$\widetilde{h}_j(x,\widehat{\theta},\widehat{\xi}_{j-1}) = h_j(x,\theta) \qquad (\text{D.9})$$

*and*

$$\|g_j(x,\widehat{\theta},\widehat{\xi}) - h_j(x,\theta)\|_2 \leqslant \min\{\lambda,\sigma_\zeta\}$$

*In addition, suppose that $\widehat{\theta},\theta,\widehat{\xi},\xi$ satisfy* (D.7) *and* (D.8). *Then it follows that for all $x \in \mathcal{X}$,*

$$\|g_i(x,\widehat{\theta},\widehat{\xi}) - h_i(x,\theta)\|_2 \leqslant \min\{\lambda,\sigma_\zeta\}$$

*Furthermore, for $1 \leqslant i \leqslant d-1$, we additionally have*

$$\widetilde{h}_i(x,\widehat{\theta},\widehat{\xi}_{i-1}) = h_i(x,\theta)$$

Combined, the two lemmas above allow us to inductively show that the prediction of the model is not changed whenever the perturbations are bounded by (D.6), (D.7), and (D.8). Next, we show that this translates directly to an all-layer margin lower bound.

**Lemma D.9.** *In the setting of Theorem D.6, suppose there exist norm bounds $a_0,...,a_d, b_0,...,b_{d-1}$ such that whenever $\|\widehat{\theta}_i - \theta_i\|_2 \leqslant a_i$ and $\|\widehat{\xi}_i - \xi_i\|_2 \leqslant b_i$, $|F(x,\theta,\xi) - F(x,\widehat{\theta},\widehat{\xi})| < 1$ for all $x \in \mathcal{X}$. Then we obtain the following lower bound on the all-layer margin, for all $x \in \mathcal{X}$:*

$$\rho_F((\theta,\xi),x,\mathbb{1}(F(x,\theta,\xi)\geqslant 0)) \geqslant \min\{a_0,...,a_d,b_0,...,b_{d-1}\}$$

*The same lower bound applies if we consider models that use layer-based weight sharing, defined by $F'(x,\theta') \triangleq F(x,\tau^{(1)}(\theta'),\tau^{(2)}(\theta'))$ for valid weight-tying mappings $\tau^{(1)}$, $\tau^{(2)}$ (Definition D.5).*

We can combine these steps to formally complete the proof of Theorem D.6.

*Proof of Theorem D.6.* Assuming the perturbation bounds (D.6) (D.7), and (D.8) hold, we can apply induction with Lemma D.7 as the base case and Lemma D.8 as the inductive step to conclude that $|F(x,\widehat{\theta},\widehat{\xi}) - F(x,\theta,\xi)| \leqslant \sigma_\zeta < 1$ for all $x \in \mathcal{X}$. We can now apply Lemma D.9 to obtain the desired bound on the all-layer margin. $\square$

We fill in the proofs of the supporting lemmas below.

*Proof of Lemma D.7.* By our definitions and Condition D.3, we have

$$\|g_0(x,\widehat{\theta},\widehat{\xi}) - h_0(x,\theta)\|_2 = \|f_0(x,\widehat{\theta}_0) - f_0(x,\theta_0)\|_2 \leqslant \mu_0\|\theta_0 - \widehat{\theta}_0\|_2 \leqslant \min\{\lambda,\sigma_\zeta\}$$

Now we can apply the Definition D.1 of the correction function to get

$$\widetilde{h}_0(x,\widehat{\theta},\xi) = \zeta_0(g_0(x,\widehat{\theta},\widehat{\xi}),\xi_0) = h_0(x,\theta)$$

$\square$

*Proof of Lemma D.8.* By expanding the expression for $h_i$, we observe that

$$\begin{aligned}
h_i(x,\theta) &= f_i(h_0(x,\theta),...,h_{i-1}(x,\theta),\theta_i) \\
&= f_i(\widetilde{h}_0(x,\widehat{\theta},\xi),\widetilde{h}_1(x,\widehat{\theta},\widehat{\xi}_0)...,\widetilde{h}_{i-1}(x,\widehat{\theta},\widehat{\xi}_{i-2}),\theta_i)
\end{aligned} \tag{D.10}$$

We obtained the equality via (D.9). Now we write

$$g_i(x,\widehat{\theta},\widehat{\xi}) = f_i(\widetilde{h}_0(x,\widehat{\theta},\widehat{\xi}),...,\widetilde{h}_{i-1}(x,\widehat{\theta},\widehat{\xi}),\widehat{\theta}_i) \tag{D.11}$$

We subtract the two expressions and add and subtract $f_i(\widetilde{h}_0(x,\widehat{\theta},\xi),\widetilde{h}_1(x,\widehat{\theta},\xi_0)...,\widetilde{h}_{i-1}(x,\widehat{\theta},\xi_{i-1}),\widehat{\theta}_i)$ to obtain

$$g_i(x,\widehat{\theta},\widehat{\xi}) - h_i(x,\theta) = E_1 + E_2$$

where

$$\begin{aligned}
E_1 &\triangleq f_i(\widetilde{h}_0(x,\widehat{\theta},\widehat{\xi}),...,\widetilde{h}_{i-1}(x,\widehat{\theta},\widehat{\xi}),\widehat{\theta}_i) \\
&\quad - f_i(\widetilde{h}_0(x,\widehat{\theta},\xi),\widetilde{h}_1(x,\widehat{\theta},\widehat{\xi}_0)...,\widetilde{h}_{i-1}(x,\widehat{\theta},\widehat{\xi}_{i-2}),\widehat{\theta}_i) \\
E_2 &\triangleq f_i(\widetilde{h}_0(x,\widehat{\theta},\xi),\widetilde{h}_1(x,\widehat{\theta},\widehat{\xi}_0)...,\widetilde{h}_{i-1}(x,\widehat{\theta},\widehat{\xi}_{i-2}),\widehat{\theta}_i) \\
&\quad - f_i(\widetilde{h}_0(x,\widehat{\theta},\xi),\widetilde{h}_1(x,\widehat{\theta},\widehat{\xi}_0)...,\widetilde{h}_{i-1}(x,\widehat{\theta},\widehat{\xi}_{i-2}),\theta_i)
\end{aligned}$$

We first bound $E_1$. We note that for all $0 \leqslant j \leqslant i-1$

$$\begin{aligned}
\|\widetilde{h}_j(x,\widehat{\theta},\widehat{\xi}) - \widetilde{h}_j(x,\widehat{\theta},\widehat{\xi}_{j-1})\|_2 &= \|\zeta_j(g_j(x,\widehat{\theta},\widehat{\xi}),\widehat{\xi}_j) - \zeta_j(g_j(x,\widehat{\theta},\widehat{\xi}),\xi_j)\|_2 \\
&\leqslant \kappa_\xi \max\{\|g_j(x,\widehat{\theta},\widehat{\xi})\|_2,1\}\|\widehat{\xi}_j - \xi_j\|_2
\end{aligned}$$

The last inequality used Condition D.3 and $\|\widehat{\xi}_j - \xi_j\|_2 \leqslant \sigma_\xi$. Now defining $H' \triangleq (\widetilde{h}_0(x,\widehat{\theta},\widehat{\xi}),...,\widetilde{h}_{i-1}(x,\widehat{\theta},\widehat{\xi}))$ and $H \triangleq (\widetilde{h}_0(x,\widehat{\theta},\xi),\widetilde{h}_1(x,\widehat{\theta},\widehat{\xi}_0)...,\widetilde{h}_{i-1}(x,\widehat{\theta},\widehat{\xi}_{i-2}))$, it follows that

$$\|H - H'\| = \max_{0 \leqslant j \leqslant i-1} \kappa_\xi \max\{\|g_j(x,\widehat{\theta},\widehat{\xi})\|_2,1\}\|\widehat{\xi}_j - \xi_j\|_2$$

Plugging in $\|g_j(x,\widehat{\theta},\widehat{\xi})\|_2 \leqslant \|h_j(x,\theta)\|_2 + \|g_j(x,\widehat{\theta},\widehat{\xi}) - h_j(x,\theta)\|_2 \leqslant 2\lambda$, $\lambda \geqslant 1$, and $\|\widehat{\xi}_j - \xi_j\|_2 \leqslant \frac{\sigma_h}{2\kappa_\xi\lambda}$, we obtain $\|H - H'\| \leqslant \sigma_h$. Furthermore, we note that $H \in (h_0, ..., h_{i-1})(\mathcal{X})$, so we can apply Condition D.3 and Definition D.2 to obtain

$$\begin{aligned}
\|E_1\|_2 &= \|f_i(H',\widehat{\theta}_i) - f_i(H,\widehat{\theta}_i)\|_2 \\
&\leqslant \mu\|H - H'\| \qquad\qquad \text{(since } \|\widehat{\theta}_i - \theta_i\|_2 \leqslant \sigma_\theta \text{ and } \|H - H'\| \leqslant \sigma_h) \\
&\leqslant 2\lambda\mu\kappa_\xi \max_j\|\widehat{\xi}_j - \xi_j\|_2
\end{aligned}$$

Next, we bound $E_2$ by applying Condition D.3 and Definition D.2 again, using $\|\widehat{\theta}_i - \theta_i\|_2 \leqslant \sigma_\theta$:

$$\begin{aligned}
\|E_2\|_2 &= \|f_i(H,\widehat{\theta}_i) - f_i(H,\theta_i)\|_2 \\
&\leqslant \kappa_\theta \|\widehat{\theta}_i - \theta_i\|_2 \max\{\|\|H\|\|,1\} \\
&= \kappa_\theta \|\widehat{\theta}_i - \theta_i\|_2 \max\{\|h_j(x,\theta)\|_2\}_j \cup \{1\} \\
&\leqslant \kappa_\theta \|\widehat{\theta}_i - \theta_i\|_2 \lambda
\end{aligned}$$

where we applied Condition D.4. By triangle inequality, follows that

$$\begin{aligned}
\|g_i(x,\widehat{\theta},\widehat{\xi}) - h_i(x,\theta)\|_2 &\leqslant \|E_1\|_2 + \|E_2\|_2 \\
&\leqslant \kappa_\theta \|\widehat{\theta}_i - \theta_i\|_2 \lambda + 2\lambda\mu\kappa_\xi \max_j \|\widehat{\xi}_j - \xi_j\|_2
\end{aligned}$$

Now by the assumptions on $\|\widehat{\theta}_i - \theta_i\|_2$ and $\|\widehat{\xi}_j - \xi_j\|_2$, we can check that the r.h.s. is bounded by $\min\{\lambda,\sigma_\zeta\}$.

Finally, we note that by Definition D.1 of the correction function, we have

$$\widetilde{h}_i(x,\widehat{\theta},\widehat{\xi}_{i-1}) = \zeta_i(g_i(x,\widehat{\theta},\widehat{\xi}),\xi_i) = h_i(x,\theta)$$

where we used the fact that $\|g_i(x,\widehat{\theta},\widehat{\xi}) - h_i(x,\theta)\|_2 \leqslant \sigma_\zeta$. $\qquad\square$

*Proof of Lemma D.9.* Note that if $\|(\theta,\xi) - (\widehat{\theta},\widehat{\xi})\|_2 < \bar{a} \triangleq \min\{a_0,...,a_d,b_0,...,b_{d-1}\}$, then by the conditions of the lemma, $|F(x,\theta,\xi) - F(x,\widehat{\theta},\widehat{\xi})| < 1$. However, because $F(x,\theta,\xi) \in \{-1,+1\}$ for all $x \in \mathcal{X}$, the sign of the output is unchanged, which means $F(x,\theta,\xi)F(x,\widehat{\theta},\widehat{\xi}) > 0$. This means that we must perturb $(\theta,\xi)$ by $\|\cdot\|_2$-norm at least $\bar{a}$ to satisfy the constraint in the all-layer margin definition, giving us the lower bound. We note that a similar argument applies to layer-based weight sharing because there are no parameters shared within a layer, so if the perturbation to $\theta'$ has $\ell_2$ norm less than $\bar{a}$, the parameters in $\tau^{(1)}(\theta'), \tau^{(2)}(\theta')$ will also have a perturbation of at most $\bar{a}$ in each layer. The same reasoning as before then applies. $\qquad\square$