# OpenReview forum: "Statistically Meaningful Approximation: a Case Study on Approximating Turing Machines with Transformers"
_NeurIPS.cc/2022/Conference — NeurIPS 2022 Accept_

### Official Review · Reviewer_FoEE · 2022-07-11

**Rating:** 7
**Confidence:** 4
**Soundness:** 4 excellent
**Presentation:** 3 good
**Contribution:** 4 excellent

**Summary:**

This paper presents an alternative to expressivity (which is muddied by issues like unbounded precision), SM-approximation, which combines both expressivity and generalization. It then shows that FFNNs SM-approximate Boolean circuits and Transformers SM-approximate bounded-time Turing machines.



**Questions:**

lines 145-149. I was troubled by the appearance of the number 0.99 in a formal definition. Would there be any negative consequence of using 1-\delta instead?

line 186: How hard is it to compute \rho_F? What would the implications be if it were very hard?


**Limitations:**

I did not see an explicit discussion of weaknesses.

There isn't a statement about potential negative societal impact, and I don't think one is needed.

**Strengths And Weaknesses:**

# Strengths

The new definition of SM-approximation is reasonable. I think the paper makes a good case for considering expressivity and generalization together.

# Weaknesses

There are no negative results, e.g., “function class \mathcal{G} is not SM-approximable by FFNNs,” or "if function class \mathclass{G} has (some property), it is not SM-approximable by FFNNs." To me such results would be equally interesting, if not more interesting. This is mentioned in the future work section, but I don't have a good sense for how difficult it would be to obtain such results. I don't consider lack of such results a weakness in this paper, but if the definition of SM-approximation made it infeasible to obtain such results, I'd consider that a major weakness of the definition.

# Minor Comments

lines 133-159. Please state that functions in \mathcal{F}, \mathcal{G} are from \mathcal{X} to \mathbb{R}.

line 186: \min_\delta

line 326-265. There are a number of assumptions made about the Transformer, most of which are very mild.

The paper claims that the following could be dispensed with in favor of standard Transformers:
- hardmax attention (distributing attention equally among ties)
- no residual connections

The paper makes no claim about:
- positional encodings are binary representation of integer position
- no layer normalization
- FFNNs have 3 layers and the last one also has a ReLU

The only important fact is:
- the number of layers is O(log T)

---

> ### Author Response · Authors · 2022-08-02
> **Response**
>
> We thank the reviewer for the insightful review and for noting that “the paper makes a good case for considering expressivity and generalization together”. We will incorporate minor comments into the next draft and respond to the main points below.
>
> — “There are no negative results, e.g., “function class $\mathcal{G}$ is not SM-approximable by FFNNs,” … I don't have a good sense for how difficult it would be to obtain such results.”
>
> We thank the reviewer for this insightful point about proving negative results, which we agree is a very interesting direction to pursue. In general, negative results in approximation theory can be quite challenging to obtain because they require arguing that **all possible** constructions cannot work, as opposed to positive results, which only require the existence of a single good construction.
>
> However, one exciting aspect of the SM-approximation framework is that it provides a possible avenue towards obtaining negative results. If we were considering the standard notion of approximation, and not SM-approximation, it appears to be very hard to prove negative results for circuits and Turing machines without solving long standing open questions in theoretical computer science. For example, we do not even know of a good separation between depth-4 threshold circuits and logarithmic depth circuits, which means, for example, that it would be very challenging to prove negative results on the ability of neural networks to approximate (in the standard sense) boolean circuits. With the SM-approximation framework, we require a stronger notion of approximation, which means obtaining negative results may be easier. For example, one interesting and plausible negative result to try to prove would be that limited-width RNNs cannot SM-approximate Turing machines when the surrogate loss corresponds to some complexity measure such as parameter norm or all-layer margin.
>
> — “ I was troubled by the appearance of the number 0.99 in a formal definition. Would there be any negative consequence of using 1-\delta instead?”
>
> We chose to use $0.99$ instead of $\delta$ for simplicity: this allows us to focus on the approximation error $\epsilon$ in the definition of $\epsilon$-SM approximation. Compared to $\epsilon$, the failure probability $\delta$ is a less important quantity, and so we chose to replace it with the constant 0.99. In all special cases studied in this paper (circuits and TMs), if we explicitly quantified the failure probability $\delta$, the sample complexity would scale with $\log(1/\delta)$, as is standard.
>
> — “How hard is it to compute \rho_F? What would the implications be if it were very hard?”
>
> $\rho_F$ is likely computationally intractable to compute exactly (see [51] for more discussion). However, in practice, it is possible to certify that $\rho_F$ is reasonably large (which implies a good generalization bound): [51] shows that for broad classes of neural networks, it is possible to lower bound $\rho_F(\theta,x,y)$ in terms of computable quantities such as the Jacobian norms, hidden layer norms, and output margin of the network evaluated at example $x$. In practice, these lower bounds often scale inverse polynomially with depth [51], instead of inverse exponentially with depth (which is the case for the naive lower bound based on the output margin and Lipschitzness of the network: $\gamma/Lip(F)$, where the Lipschitz constant is crudely bounded by the product of some parameter norm).
>
> Furthermore, [51] also shows that an algorithm based on heuristically optimizing the all-layer margin surrogate loss defined in (2.2) can improve generalization performance in practice, which provides further justification for theoretically studying this surrogate loss in this work.
>
> — “I did not see an explicit discussion of weaknesses.”
>
> We thank the reviewer for pointing this out and will include clearer discussion of our limitations for the next revision. One potential limitation is that the “correction function” machinery discussed in Lemma 3.3 relies on the discrete nature of boolean circuits and Turing machines, and so additional work and insight would be required to prove analogous SM-approximation results for continuous functions (please see our discussion with Reviewer i8t1 for speculation on extending our analysis to continuous functions).

---

### Official Review · Reviewer_k6X3 · 2022-07-11

**Rating:** 7
**Confidence:** 2
**Soundness:** 4 excellent
**Presentation:** 4 excellent
**Contribution:** 3 good

**Summary:**

The paper introduces the novel notion of statistically meaningful approximation, which requires that a function family can be approximated with high probability based on the empirical risk minimizer obtained from $n$ samples. This notion combines both approximation and generalization and does not allow for infinite precision (as used by prior work), which can inflate the expressivity of an architecture. The paper demonstrates that feedforward neural networks can statistically meaningfully approximate boolean circuits and that transformers can statistically meaningfully approximate Turing machines.

**Questions:**

Are the sequences in $\mathcal{A}^\ast$ on line 127 of finite length?

**Limitations:**

The paper does not address limitations or potential negative societal impacts.

**Strengths And Weaknesses:**

**Strengths**

The paper introduces a novel notion of approximation and generalization for neural networks, which can potentially open up a fruitful line of future research. Indeed, the paper shows that this notion of statistically meaningful approximation allows for improved bounds over prior work for approximating boolean circuits with feedforward neural networks and for approximating bounded-time Turing machines with transformers. Concretely, the paper improves the sample complexity from exponential depth scaling and polynomial parameter scaling to scaling in the circuit size for the approximation of boolean circuits. Similarly, the paper presents an exponential scaling improvement for approximation of Turing machines with bounded computation time $T$ (sample complexity polynomial in $\log T$ instead of $T$). Importantly, statistically meaningful approximation does not allow for infinite precision, which is known to inflate the expressivity of neural network architectures. Finally, the paper is very well-written and well-placed in the literature.

**Weaknesses**

The definition of the 0-1 loss on line 181 is incorrect. Consider, e.g., $y = z = 0$, which corresponds to a loss of $0$ but yields $\mathbb{1}((0 - 0.5)0 \leq 0) = \mathbb{1}(0 \leq 0) = 1$. Similarly, the definition of the all-layer margin is incorrect (e.g., $y = 0$ and $F(x, \theta) = 0$ yields $\rho = 0$, which is not strictly positive). Since all other proofs in the paper hinge on this definition, I do not know whether they are also incorrect.


I am happy to increase my score if the above concerns are addressed.

---

> ### Author Response · Authors · 2022-08-02
> **Response**
>
> We thank the reviewer for the overall positive review which notes that our paper is “very well-written and well-placed in the literature”. We can address the reviewer’s concern expressed in the “weaknesses” section and hope that the reviewer will consider increasing their score after reading this response.
>
> — Addressing the reviewer’s main concern: “definition of the 0-1 loss on line 181 is incorrect. Consider, e.g., $y=z=0$, which corresponds to a loss of 0 but yields $1((0−0.5)0\le 0)=1(0 \le 0)=1$. Similarly, the definition of the all-layer margin is incorrect (e.g., $y=0$ and $F(x,\theta)=0$ yields $\rho=0$, which is not strictly positive).”
>
> Thanks for the question! This is indeed the desired behavior, and not a mistake, though we will be sure to clarify in the next version of the paper. We suspect that the confusion stems from the fact that we use notations that are slightly different from those the reviewer is familiar with. Note that $y$ corresponds to the target label and belongs to $\{0,1\}$, whereas $z$ is the margin predicted by the model and is a real number. The binary prediction of the model will be $\mathbf{1}(z\ge 0)$. Therefore, for a prediction to be “correct”, we require $y$ to match $\mathbf{1}(z\ge 0))$, or in other words, we require $z$ to be **negative** when $y = 0$, and positive when $y = 1$. One can verify that our definition of the loss $\mathbf{1}((y-0.5)z\le 0)$ gives this behavior. Regarding the specific setting of $y=z=0$ that the reviewer brought up, we note that $z = 0$ corresponds to an ambiguous prediction, and could therefore be penalized by the loss. (An alternative way of writing the loss is $\mathbf{1}(y = \mathbf{1}(z\ge 0))$. It’s equivalent to our definition except for the boundary case when $z=0$. The loss $\mathbf{1}(y = \mathbf{1}(z\ge 0))$ considers $z=0$ as corresponding to outputting label 1, whereas our loss penalizes $z=0$ in all cases because it’s the completely ambiguous output.)
>
> If we use the convention that $y \in \{-1, +1\}$, then the loss function will be $\mathbf{1}(yz \le 0)$, which is perhaps more familiar in the margin literature.  We will consider using this notation in the future if it’s clearer.
>
> — “Are the sequences in $A^*$ on line 127 of finite length?”
>
> Yes, in particular, sequences in $\mathcal{A}^*$ can be assumed to have length at most $T$, as this is the maximum computation time of the Turing machine, and thus the maximum amount of input symbols the Turing machine can possibly read.

---

> > ### Comment · Reviewer_k6X3 · 2022-08-09
> > **Thank You for the Clarification**
> >
> > Thank you for clarifying my concerns! I have raised my score to reflect my positive opinion of the paper, given that the authors incorporate the above clarifications.

---

### Official Review · Reviewer_i8t1 · 2022-07-14

**Rating:** 7
**Confidence:** 3
**Soundness:** 3 good
**Presentation:** 3 good
**Contribution:** 4 excellent

**Summary:**

This paper studies the approximation power of neural networks under the constraint of finite-precision weights and activations. Many classical results touting the expressive power of neural networks have a strong reliance on infinite precision weights and activations whereas the more practical finite-precision networks are nearly not as expressive.

In this paper, authors propose a different take on how one should evaluate expressivity. Instead of focusing on whether a target function can be expressed by the class of neural networks in question, one should focus on the statistical properties exhibited by the class. To that effect, authors propose the notion of statistically meaningful (SM) approximation which looks at how well ERM can approximate a given function.

Under this notion of approximation, authors first show  that neural networks can approximate boolean circuits where the sample complexity only depends on the polynomial of the circuit size and only have a logarithmic dependence on the parameters of the neural network. Secondly, authors show that that with the transformers can approximate turning machine that run for at most $T$ steps.

The main technical contribution of this paper is the proof technique used by the authors to derive the approximation results.

The first main contribution is the all-layer margin loss that allows one to reason about finite precision weights.

Second contribution is Lemma 2.4 that, at a high level, says that for any $G\in \mathcal{G}$ one can find a function $F$ in the approximating class $\mathcal{F}$ with large all-layer margin approximates , then the $\mathcal{F}$  $\epsilon$-SM approximates $\mathcal{G}$ with sample complexity depends on the margin and on $\log$ of the number of parameters.

As a high-level proof technique for both boolean circuits and Turing machines, authors first showed by construction a neural network that approximate a given boolean circuit/Turning machine. To get the appropriate sample complexity result, the approximating function needs large margin. To get large margin, authors introduce “correction” functions that needs to be added to the constructed neural network. Due to the discrete nature of operations in both TM and boolean circuits, the correction functions are implemented by $\text{round}$ operations.

**Questions:**

I am not entirely sure how the reading operation at location $l_i(x)$ is achieved. How can one binary search over the past steps when the $l_j(x)$ for $j\leq i$ are not sorted?

**Limitations:**

Authors have addressed limitations of their work.

**Strengths And Weaknesses:**

**Strengths:**

- This is a well motivated paper that studies the expressivity of neural networks from a statistical lens.
- To the best of my knowledge, the technique proposed in the paper is novel and the ideas are original which can be applied to study approximation power of different neural network architecture.
- Lemma 2.4 also has implications for studying generalization properties of neural networks where all-layer margin loss depends on $\ell_2$ stability of the neural network.

**Weaknesses:**

- The construction of self-attention network to simulate Turing machine is a bit hard to follow from the body of the paper. It’s still not clear to me how the reading at location $l_i$ is achieved by the transformers.
- As far as I understand, authors are able to construct correction functions due to the discrete nature of boolean circuits and TM. Constructing for other class of functions is probably more challenging thus it’ll be difficult to adapt this proof technique.

---

> ### Author Response · Authors · 2022-08-02
> **Response**
>
> We thank the reviewer for the thorough and encouraging review and for noting that “the technique proposed in the paper is novel” and our work has “implications for studying generalization properties of neural networks”.
>
> — “how the reading operation at location $l_i(x)$ is achieved. How can one binary search over the past steps when the $l_j(x)$ for $j \le i$ are not sorted?”
>
> Define $i’$ as in line 404 of the paper. The key idea is to search over **bits** of $i’$. For example, roughly speaking, the first step in the binary search will determine whether $i’$ is in the first [0, T/2] timesteps or last [T/2, T]. A single transformer layer (attention layers followed by feedforward layers) can implement one step in this binary search, and therefore the entire binary search can be implemented via $O(\log T)$ transformer layers.
>
> We now provide a rough sketch of a single step of the binary search, though for more details we refer the reviewer to Claim C.5 and the surrounding text. Note that at the $(j + 1)$-th step of the search, we maintain the invariant that the first $j$ bits of $i’$ are already recorded. Now the idea is to construct the attention layers such that 1) we only attend to timesteps where the first $j$ bits match those of $i’$, and 2) amongst such timesteps, if there is a timestep $t$ with $(j + 1)$-th bit 1 and $l_t(x) = l_{i + 1}(x)$, the attention score is maximized. For example, property 1) can be implemented by making the keys and query record the first $j$ bits of the timestep $t$ and $i’$, respectively, such that the <key, query> dot product is highest when the first $j$ bits match. Property 2) can similarly be implemented by recording $l_t(x)$ and $l_{i + 1}(x)$ in the time step $t$ keys and query vectors. (More details are in Claim C.5). By combining these properties, we can uncover the $(j + 1)$-th bit of $i’$.
>
> — “authors are able to construct correction functions due to the discrete nature of boolean circuits and TM. Constructing for other class of functions is probably more challenging”
>
> Towards obtaining a first-cut result, we indeed leveraged the discrete nature of boolean circuits and TMs in relying on the correction function machinery to enforce a large all-layer margin. Analyzing other class of functions is an interesting direction for future work. One important property of discrete functions, which we suspect may be leveraged more generally, is that it is easy to correct errors in intermediate computations of discrete functions (by rounding). It would be interesting to see whether this property has a continuous analog which can be analyzed.

---

### Official Review · Reviewer_AqPm · 2022-07-15

**Rating:** 6
**Confidence:** 4
**Soundness:** 3 good
**Presentation:** 2 fair
**Contribution:** 2 fair

**Summary:**

This paper introduces a notion of statistical meaningful approximation and applies it to studying how NN models can approximate Boolean circuit and Turing Machines. One aspect of their definition is using two families of functions: One family F (e.g., NN) approximating a second family (e.g., Turing machines). Various generalization bounds are given with respect to this new notion.

**Questions:**

"finite width RNNs with infinite precision can simulate Turing machines,
but finite-precision, finite-width RNNs cannot, as implied by streaming lower bounds"

Can you prove this statement of back it up with a paper proving it?

**Limitations:**

With many parts of the paper the feeling is that authors are using sub optimal phrasing and references.
"Unfortunately, a rigorous analysis of optimization is unresolved even for simple two-layer nets"
The reference for this is very strange. There are many much more relevant works I would cite.

The discussion of infinite precision also does not mention the large body of work dealing with weight-restricted NN as well
as results showing how to avoid large weights by increasing depth. And there are more such examples. I would probably cite
"For valid generalization the size of the weights is more important than the size of the network"

There are many other improvements: for example replace "boolean" by Boolean. I recommend a though pass improving the language and the presentation.

**Strengths And Weaknesses:**

Long story short I find this paper to be weak accept: I support acceptance provided the authors can demonstrate they can deal with the (many) issues in this paper.

I'm less convinced by the SM definition. To me it feels similar to other notions of learnability and also the choice of a specific loss function as opposed to a family of loss functions (which seems to be necessary for the definition to go through) is another disadvantage.
Rather, the strength of the paper is in the simulation results of Turing machines and Boolean circuit by NN models. These simulation results are needed and could find multiple uses.

---

> ### Author Response · Authors · 2022-08-02
> **Response**
>
> We thank the reviewer for the insightful comments and for noting that the paper’s “[approximation theory] results are needed and could find multiple uses”. We will incorporate the reviewer’s suggestions on citations into the next draft of this paper. Responses to specific points below:
>
> — “less convinced by the SM definition … feels similar to other notions of learnability and also the choice of a specific loss function as opposed to a family of loss functions … is another disadvantage”
>
> The main purpose of the SM-approximation definition is to pin down a concrete and mathematical notion of “meaningful” approximation. This addresses an important issue in the field of approximation theory, since, as discussed in our introduction, many existing constructions are not “meaningful” because of their reliance on infinite precision.
>
> There are indeed similarities between SM approximation and other notions of learnability, but, as discussed in lines 153-159, there is a crucial conceptual difference: concepts such as PAC-learning, agnostic learning, and improper learning are mainly concerned with generalization in learning the target class $\mathcal{G}$. In contrast, in SM approximation, it is also nontrivial to show that $\mathcal{F}$ is expressive enough to approximate functions in $\mathcal{G}$.
>
> Regarding single loss functions v.s. families of losses: we assume the reviewer is referring to defining approximation w.r.t. the single loss function $\ell$ as opposed to some family $\mathcal{L}$. It is unclear to us that achieving low approximation error for a family of loss functions is more desirable than low approximation error for a single loss $\ell$. For example, when $\ell$ is a single function such as squared loss or 0-1 loss (as studied in this paper), achieving 0 loss means that $\mathcal{F}$ perfectly expresses the target class $\mathcal{G}$.
>
> — “‘finite width RNNs with infinite precision can simulate Turing machines, but finite-precision, finite-width RNNs cannot, as implied by streaming lower bounds’ Can you prove this statement or back it up with a paper proving it?”
>
> Any finite-precision, finite-width RNN induces a finite-space streaming algorithm corresponding to simply running the RNN on the inputs. However, streaming lower bounds tell us that finite-space streaming algorithms are not powerful enough to simulate Turing machines, and hence finite-precision, finite-width RNNs cannot either. (See [1] referenced in the paper for an example of a streaming lower bound).
>
> — “‘Unfortunately, a rigorous analysis of optimization is unresolved even for simple two-layer nets’ The reference for this is very strange. There are many much more relevant works I would cite.”
>
> We thank the reviewer for this comment – we will update references in the next draft of the paper. As an aside, we’d also like to clarify that there do exist global optimization analyses such as the NTK line of work, but these analyses cannot sufficiently account for the good generalization of neural nets (there is a large body of theoretical and empirical work showing that neural networks can generalize much better than NTK analyses can hope to prove; see e.g. [Ghorbani et al.’19, Wei et al.’19]). There is also some limited progress on analyzing the optimization of neural networks beyond the NTK training regime. We will revise the phrasing in the next draft to include more discussion of these nuances.
>
> References:
>
> Ghorbani, Behrooz, Song Mei, Theodor Misiakiewicz, and Andrea Montanari. "Limitations of lazy training of two-layers neural network."
> Wei, Colin, Jason D. Lee, Qiang Liu, and Tengyu Ma. "Regularization matters: Generalization and optimization of neural nets vs their induced kernel."

---

> > ### Comment · Reviewer_AqPm · 2022-08-08
> > **Comment**
> >
> > " (See [1] referenced in the paper for an example of a streaming lower bound)."
> >
> > I feel that this is not sufficient. It would be beneficial to refer to a specific problem and explain why does the lower bound apply to RNNs as well as state what the lower bound is.

---

> > > ### Author Response · Authors · 2022-08-09
> > > **Response**
> > >
> > > Thanks for the comment. For example, the streaming lower bound on approximating frequency moments in [1] could be directly plugged into our earlier statement "streaming lower bounds tell us that finite-space streaming algorithms are not powerful enough to simulate Turing machines, and hence finite-precision, finite-width RNNs cannot either." As suggested by the reviewer, we will include discussion with this specific problem in the paper.

---

### Meta-Review · Area_Chair_E9Kq · 2022-08-28

**Recommendation:** Accept
**Confidence:** Certain

**Metareview:**

Reviewers agree on the merits of sharing the paper with the community.The authors are highly encouraged to incorporate the many constructive suggestions offered.

**Award:**

No

---

### Decision · Program_Chairs · 2022-09-14

Accept